# Response Time to Flood Events using a Social Vulnerability Index (ReTSVI)

Alvaro Quezada-Hofflinger[1], Marcelo A. Somos-Valenzuela[2,3], Arturo Vallejos-Romero[1]

[1]Nucleo de Ciencias Sociales, Universidad de la Frontera, Temuco, 4780000, Chile

[2]Agricultural and Forestry Sciences, Universidad de La Frontera, Temuco 4780000, Chile

[3] Northeast Climate Science Center, University of Massachusetts, Amherst, 01003, USA

*Correspondence to*: Marcelo A. Somos-Valenzuela (marcelo.somos@ufrontera.cl)

**Abstract.** Current methods to estimate evacuation time during a natural disaster assume that human responses across different social groups are similar. However, individuals respond differently based on
their socioeconomic and demographic characteristics. This article develops the Response Time by Social Vulnerability Index (ReTSVI). ReTSVI combines a series of modules which are pieces of information that interact during an evacuation, such us evacuation rate curves, mobilization, inundation models and social vulnerability indexes to create an integrated map of evacuation rate in a given location. We provide an example of the application of ReTSVI in a potential case of a severe flood
event in Huaraz, Peru. The results show that during the first 5 minutes of the evacuation, the population that lives in neighbourhoods with high social vulnerability evacuate 15% and 22% fewer people than the blocks with medium and low social vulnerability. These differences gradually decrease over time after the evacuation warning, and social vulnerability becomes less relevant after 30 minutes although, with the data available, with not statistical significance. Using a methodology such as ReTSVI allows
first responders to identify areas where the same level of physical vulnerability affects distinct groups differently.

Keywords: ReTSVI, Social Vulnerability, Flood Hazard Evacuation

# 1 INTRODUCTION

The costs associated with health, food security and the physical environment produced by climate change are expected to reach between 2 and 4 trillion US dollars by 2030 (Hallegatte, 2014). The United Nations has indicated that the frequency and severity of climate change-related natural disasters
are expected to increase faster than risk reduction can be achieved (UN, 2009). For example, worldwide natural disasters caused around 3.5 trillion US dollars in damages from 1980 5 to 2011, one-third took place in low or middle-income countries, and the number of people affected by natural disasters increased 1.5 times, economic damage by 1.8 times and total deaths by two times (Basher, 2006; Hallegatte, 2014). A key strategy to reduce the loss of human life during a disaster is to improve preparedness of
communities. A common means to achieve this is to develop Early Warning Systems (EWS) to alert the population to evacuate before disaster strikes. Ideally, EWS should consider not only the so called physical dimensions such as exposure and intensity, but also the human or social dimensions that help us to understand differences in response to similar stresses (Basher, 2006; Bouwer, 2011; Nagarajan, Shaw, & Albores, 2012; Nicholls & Klein, 1999). Individual characteristics such as race, age, gender,
education, income, and employment influence the susceptibility to which certain groups or communities might be exposed and also define their ability to respond to a natural hazard (Cutter, Boruff, & Shirley, 2003; Gaillard & Dibben, 2008). For example, women and men or those people with different levels of physical and cognitive ability, experience and respond to disasters differently (Cutter & Finch, 2008; Ionescu, Klein, Hinkel, Kavi Kumar, & Klein, 2005; ISDR, 2004; Santos & Aguirre, 2004). Despite the
evidence, the literature focuses mainly on the physical dimension of natural hazards and disregards human aspects. A real improvement in our understanding of emergency evacuations will depend on the integration of both (Basher, 2006; Couling, 2014; Santos & Aguirre, 2004).
The problem that arises is how we can incorporate social and physical vulnerability in a comprehensive matter to improve our understanding of an evacuation process. Both concepts have been developed
independently in the social sciences and engineering; therefore, it is not a straightforward process to link them. In fact, there is little data on how social vulnerability influences the evacuation process and how it is linked to the number of human casualties (Bolin, 2007; Morss, Wilhelmi, Meehl, & Dilling, 2011). To address this problem, some scholars have mapped physical and social vulnerability to visualize how they overlap. They have also combined them using arithmetic operations such as
multiplication or addition of social and physical vulnerability indexes to create a unique indicator that considers both vulnerabilities to study evacuation (Chakraborty, Tobin, & Montz, 2005)) or recovery process after hazards occur (Cutter & Emrich, 2006; Hegglin & Huggel, 2008). This information is still descriptive and provides qualitative information to policy makers, government institutions or local governments to understand how a population would react in an evacuation process. Therefore, questions
such us: what it means to live in a neighborhood with high physical and social vulnerability? and, how much time will the population need to evacuate neighborhoods with high social vulnerability and low physical vulnerability? are not possible to answer with the current methods developed in social sciences nor engineering.

## 1.1 Social Vulnerability and Natural Disasters
Recent major natural disasters such as Hurricane Katrina and the 2010 earthquake in Haiti have shown the relevance of integrating social vulnerability into risk management and decision-making (Flanagan,

Gregory, Hallisey, Heitgerd, & Lewis, 2011). This integration refers to identifying which and where problems exist before natural disaster strikes, making it possible to take steps to prevent possible damage (Schmidtlein, Deutsch, Piegorsch, & Cutter, 2008). In this context, a better understanding of how problems like segregation, socioeconomic deprivation and inequalities affect the type of response

and the degree of resiliency of communities affected by natural disasters is crucial. With this information, federal and local governments could be more effective in mitigating losses or improving the recovery of communities (Cutter & Emrich, 2006; Heinz Center, 2002). The degree to which communities and people are vulnerable to hazards is explained not only by proximity to potential natural disasters, but also social characteristics such as socioeconomic and demographic features that

could exacerbate or lessen the impact of a disaster (Chakraborty et al., 2005; Cutter, Mitchell, & Scott, 2000).

The study of vulnerability can be traced back to the early 1950s and 1960s in the field of behavioural sciences, the main objective of which was to understand the features of areas that make them either suitable to inhabit. During the 1970s, the US federal government was interested in the relationship

between social well-being and progress indicators; consequently, the connection between socioeconomic inequalities and social problems became clearer at that time (Cutter & Emrich, 2006). Today, the concept has broadened to include a more comprehensive approach that combines different areas, such as social, demographic, economic, and geographic vulnerability, but each discipline defines the concept differently (Alwang, Siegel, Jørgensen, & Tech, 2001; Balica, 2012; Birkmann, 2007). For

example, in the economic literature, vulnerability includes food security and sustainable development (Fekete, 2011; Rygel, O'sullivan, & Yarnal, 2006). In the disaster risk community, vulnerability is defined as the physical, social, and environmental factors that increase the likelihood of a community being impacted by hazards (Zhou et al, 2014). Models of social vulnerability, in this area, have been used to explain the capability of communities to face and recover from disasters (Chakraborty et al.,

25 2005).

Scholars have tried to understand whether socioeconomic and demographic characteristics of the population are relevant to understand why neighborhoods or communities respond differently during an evacuation, why some people evacuate, and others do not evacuate during disasters. The evidence about evacuations during hurricanes shows mixed results. Huang, Lindell, & Prater (2016) analyzed 49

studies linked to evacuations to hurricane warnings conducted since 1991 and concluded that demographics variables have a minor or inconsistent impact on household evacuations. In contrast, others studies show that social vulnerability is a key factor to take into account during emergency management and evacuation planning (Bateman & Edwards, 2002; Chakraborty et al., 2005; Dash & Gladwin, 2007; Kusenbach, Simms, & Tobin, 2010). In the case of floods, studies suggest that social

vulnerability is an important element to consider in order understanding different behaviors during flooding evacuations. In particular, scholars have found that variables such as low household income, poor housing quality, children (Pelling, 1997), women, housewives, students (De Marchi, 2007), elderly, high population density and population with low level of education (Zhang & You, 2014) are key variables to consider to create a social vulnerability index linked to evacuations during disasters.

Research in social vulnerability linked to natural hazards can be divided into two groups. The first group, "post-disaster cases studies," tries to understand how natural disasters impact differently communities based on their level of social vulnerability (Rufat, Tate, Burton, & Maroof, 2015). Most of

the research in this area uses qualitative methods, such as semi structured interviews, focus groups, key informant interviews and participant observation (Działek, Biernacki, Fiedeń, Listwan-Franczak, & Franczak, 2016). One of the main limitations of these studies is that their findings cannot be generalized to aggregated levels such as regions or countries. The second group of research is in "geospatial

modelling studies." Scholars in this subfield use primarily quantitative methods and focus on creating maps or developing indexes to compare the different levels of social vulnerability among communities, regions or countries (Rufat et al., 2015). A central aim of developing techniques to quantify vulnerability is to reduce gaps between theoretical concepts of vulnerability and the decision-making process (Birkmann, 2007).

There are multiple challenges in constructing an index to measure the social vulnerability of a certain population. The most evident is the degree of subjectivity in the selection of variables as well as the application and operationalization of vulnerability as a concept (Fekete, 2011). Furthermore, an index does not indicate the structure and causes of social vulnerability; therefore, using a single factor to measure vulnerability might disregard the importance of particular variables that are relevant to

explaining social vulnerability in a particular area (Rygel et al., 2006). In fact, the capability of communities to cope with and recover from disaster seem to depend also from other factors such as vigor, vitality, energy, strength, etc., which are usually excluded from studies about social vulnerability (De Marchi, 2007; De Marchi & Scolobig, 2012). Despite these limitations, scholars have developed indices to quantify social vulnerability based on their interests. Some researchers use the percentage of

women, racial groups or age average as indexes to estimate different levels of social vulnerability (Harvey, Kato, & Passidomo, 2016; Sebastiaan N Jonkman, Maaskant, Boyd, & Levitan, 2009; Sadia, Iqbal, Ahmad, Ali, & Ahmad, 2016). Other scholars use variables linked with social vulnerability as independent variables in regression models (Działek et al., 2016); variables are simply ranked from lowest to highest values (Flanagan et al., 2011) or using the weighted average to estimate social

vulnerability (Adger & Vincent, 2005). However, these indexes have some limitations. Namely, they use a limited number of variables and do not consider the interrelationship among variables to quantify social vulnerability. To address this problem, researchers have employed strategies such as including a higher number of variables to construct social vulnerability indexes or estimating the connection among variables that are linked theoretically with social vulnerability. In this area, one of the most recognized

indices to have been applied both in the US and abroad is the Social Vulnerability Index (SoVI) (Cutter, 1996). SoVI has been used in California, Colorado and South Carolina in the USA, and in countries such as England, Australia, Germany, and Norway (Zhou et al., 2014). The SoVI approach has been replicated in different geographical settings, and on different spatial and temporal scales (Schmidtlein et al., 2008). The use of SoVI is relevant because the method makes it possible to compare the spatial

variability in socioeconomic vulnerability using a single index value. SoVI can also be linked spatially to physical aspects to calculate the overall vulnerability of a specific place (Boruff, Emrich, & Cutter, 2005).
Social vulnerability indexes are useful to detect differences in social vulnerability to flood events (Fekete, 2009). In particular, the Social Vulnerability Index (SoVI) (Cutter, 1996) is adaptable to

developing countries since it can be constructed using Census data from the area of study.
The literature identifies several variables that contribute to social vulnerability (post disaster). At the individual level, social vulnerability is related to poverty and health indices, age and education level. At

the community level, social vulnerability is affected by income distribution, access to economic assets, and qualitative indicators of institutional arrangements (Adger, 1999). Furthermore, Fekete (2010) identified key variables that may explain the different levels of social vulnerability such as age group, gender, income, education, whether one owns a home, social capital, and household size. Cutter, Boruff, & Shirley (2003) also included race and ethnicity, commercial and industrial development, unemployment, rural/urban residency, residential property, infrastructure and lifelines, occupation, family structure, population growth, medical services, social dependence and special needs populations as fundamental variables to quantify social vulnerability. In the case of evacuation process during hurricanes and floods, variables such as number of housing units, mobile homes, poverty, age, people with disabilities (Chakraborty et al., 2005), education, household income, pet ownership (Kusenbach et al., 2010), household size, elderly, children (Dash & Gladwin, 2007), household quality, community organization (Pelling, 1997), communities' immaterial characteristics as energy, vigour, vitality (De Marchi, 2007; De Marchi & Scolobig, 2012), average number of people per house, population density (person/km2), illiterate population and urban population ration (Zhang & You, 2014)

## 1.2 Response Time, Evacuation and Flood Impacts

Multiple factors seem to affect people's decision-making process to evacuate, such as risk perception, beliefs, demographic characteristics, previous knowledge, social networks, gender, age and class, among others (Elliott & Pais, 2006; Lindell, Lu, & Prater, 2005; Mileti & O'Brien, 1992; Whitehead et al., 2000). Understanding what factors influence people's decisions in an evacuation is relevant because this information could help to improve the evacuation process, for example, reducing the time of evacuation response, and consequently decreasing the percentage of human casualties.

The most sensitive cost of disaster is the loss of life; nonetheless, a limited number of methods estimate the loss of life due to natural disasters and just a few of them consider social vulnerability as an explanatory variable in their models (S. N. Jonkman, Vrijling, & Vrouwenvelder, 2008).

In ocean and river floods, variables such as the percentage of buildings collapsed, the proportion of evacuated people seem to influence the number of human fatalities (Vrouwenvelder & Steenhuis, 1997). Other scholars take into account the level of water depth, flow velocity, the possibilities for evacuation, flood hazard and area vulnerability (Boyd, Levitan, & van Heerden, 2005; Jonkman, 2001). In the case of dam break floods, Brown & Graham (1988) analysed 24 major dam failures and flash floods to estimate the number of lives lost as a function of time available for evacuation and the number of people at risk, they found that time available for evacuation and population size, similar results were found by DeKay & McClelland (1993). Graham (1999) proposed that fatality rates are functions of the severity of the flood, the amount of warning time and the population's understanding of the hazard. In another example, to estimate human casualties due to flood events, the US Army Corps of Engineers developed HEC-FIA. Models, in general, assume that people react the same way during an evacuation process, and do not consider that people can respond differently based on their social vulnerability. Few authors consider the characteristics of the population to estimate human causalties during a flood event. Reiter (2001) incorporated some variables linked to social vulnerability such as the number of children and elderly to estimate the loss of life during a flood event. Penning-Rowsell and colleagues (2005) consider "people vulnerability" defined by age, disability or illness using census data. A general conclusion from the literature explored is that only a few of the methods studied have systematically included social vulnerability as an explanatory variable of human fatalities during natural disasters. In

fact, Jonkman et al. (2008) reviewed 20 methods to quantify the loss of life during different types of flood events and only found that Ramsbottom and colleagues (2004) include levels of population vulnerability, and this category is based on expert judgment. Consequently, even though there is an upward trend of research that endeavours to understand how social characteristics of population
influence human response to natural disasters, academics have failed to incorporate social vulnerability into estimations of loss of life (Elliott & Pais, 2006; Rodriguez, Quarentelli, & Dynes, 2007). We argue that this is due to the lack of understanding of how social vulnerability influences the evacuation process and human casualties (Bolin, 2007; Morss et al., 2011). In fact, current methods to quantify social vulnerability allow for the classification of neighbourhoods, counties or regions from the lowest
to highest levels of vulnerability. However, using these classifications scholars or policy makers cannot predict how many people from neighbourhoods with low vulnerability will evacuate versus those who live in neighbourhoods with high vulnerability or how much time people who live in neighbourhoods with medium vulnerability will take to evacuate versus those who live in highly vulnerable areas, etc. To fill this gap in the literature, we propose the Response Time by Social Vulnerability Index
(ReTSVI), a methodology that incorporates the demographic and socioeconomic characteristics of population into the current evacuation models.

## 2.METHODS AND DATA TO ESTIMATE ReTSVI

### 2.1 Conceptual model of ReTSVI
The objective of this work is to propose a conceptual model 'The Response Time by Social
Vulnerability Index (ReTSVI)' methodology that allows for the inclusion of social vulnerability into the traditional evacuation/mobilization models and it moves away from traditional methods that combined social vulnerability and hazard magnitude by ranking in a matrix system that results in qualitative assessment. Figure 1 is a chart of ReTSVI, we use three types of input data, which are: 1) the evacuation curves, one for each level of vulnerability (high, medium and low vulnerability); 2) a model
that describes the physical hazard that the population may be exposed to, for example, the time that a flood takes to reach a populated area; and 3) demographic information such as a census data that allows us to categorize the population into different levels of social vulnerability. Then we have two intermediate models. The first one corresponds to the mobilization model that combines the evacuation curves and the inundation model. The results of this step are three maps (one for each level of
vulnerability) of the percentage of people that evacuate before the flood strikes a place. The second intermediate model is the calculation of the social vulnerability index (SVI) using the census data, which produces a map of the city in which we can classify each block by social vulnerability. Finally, we combined the results (Integration Model Figure 1) from the mobilization model and the SVI calculations to generate a map with the percentage of people that can evacuate, which considers their
social vulnerability level."

Insert Figure 1

## 2.2 Application of ReTSVI in a potential flood in Huaraz, Peru.

In 1941, the city of Huaraz was affected by a Glacier Lake Outburst Flood (GLOF) generated at Lake Palcacocha, in the Cordillera Blanca, Peru (Figure 2). The GLOF killed in the order of 2000 people and damaged infrastructure all the way from the lake, located in the Cordillera Blanca, to the Pacific Ocean (Carey, 2010; Carey, 2005; Wegner, 2014). According to new observations and data, a new GLOF could occur at this location. In fact, Lake Palcacocha has been declared in a state of emergency several times, and currently, there are initiatives to mitigate the risk by lowering the water level and installing early warning systems (EWS) to protect the population in case a GLOF occurs (HiMAP, 2014). The physical aspects of a potential GLOF have been studied extensively with the support of international agencies such as USAID, the IDB, and the government of Peru (Rivas et al. , 2015; Somos-Valenzuela, 2014; Somos-Valenzuela et al., 2016). However, the social aspects of a flood hazard have not been studied except for qualitative studies (Hegglin & Huggel, 2008; Somos-Valenzuela, 2014).

Insert Figure 2

## 2.2.1 Input Data

To produce the ReTSVI we use three types of input data (Figure 1). First, we need the evacuation curves, one for each level of social vulnerability. Ideally, the evacuation curves that we used are generated in the area of study, however, there is no data that describes how people in Huaraz evacuate after an EWS is released; therefore, we had to generate this information. Our closest available event was the tsunami triggered by an 8.3 magnitude earthquake on 16 September 2015 in Coquimbo, Chile. Second, a model describing a potential hazard is also needed, thus we use the model of a potential GLOF in Huaraz developed by Somos-Valenzuela et al. (2016). This model provides the time that people have to react before the inundation arrives. Finally, we have the 2007 Census data provided by the Ministry of Environment of Peru to create a social vulnerability map of Huaraz.

## Surveys in Coquimbo, Chile

We conducted 22 surveys with first responders to the 8.3 magnitude earthquake and tsunami that occurred on September 16, 2015, in Coquimbo, Chile. Four institutions that work directly to help the population during the evacuation process participated in this study: the navy, the police, firefighters and the emergency office from the municipality of Coquimbo. First, we contacted by phone with each institution to explain the purpose of the study and asked them if they agree to participate in the research, all of them agree. Then, a research assistant visited each institution and asked them to select at least five emergency experts to respond to our questionnaire. The main requirement was that the participants worked directly during the emergency to help people evacuate their houses. The research assistant conducted a personal interview with each participant. We asked the first responders "In your opinion and based on your experience during the tsunami of 16[th] of September. Since the evacuation alarm was active, what is the evacuation time of population who live in areas of low/medium/high social vulnerability?" They needed to estimate the average evacuation time in neighborhoods with low, medium and high social vulnerability. Then, we asked "what is the percentage of the population that evacuate in the first X minutes? (X=5, 15, 30, 45, 60)". The first responders write down the percentage of the population that evacuates their households from 0 to 5 minutes, 0 to 15 minutes, 0 to 30 minutes, 0 to 45 minutes, 0 to 60 minutes in neighborhoods with low, medium and high social vulnerability in Coquimbo. The answers were recollected into two scales: percentages and average time (in minutes).

We use the National Socioeconomic Characterization Survey (CASEN)[1] from 2015, the same year that the earthquake/tsunami occurred, to calculate a social vulnerability index at the municipality level, following the same procedure identified in the section 2.2.3. This way we were able to identify the socioeconomic and demographic characteristics of the neighborhoods with high, medium and low social
vulnerability in Chile. We incorporate this information in the survey, so the first responders could identify what neighborhood belongs to each category; all responders generate separate curves for low, medium, or high vulnerability neighborhoods.

**Census Data from Peru**
We used the 2007 national population census to quantify the social vulnerability of Huaraz, Peru. The census has 53 questions that describe the main socio demographic characteristics of the population of Peru (INE, 2015). The census data is aggregated at the block level, and in the case of Huaraz provides full information on 1,404 blocks. The census data is divided into three main categories: (a) location of household (blocks), (b) household characteristics: number of rooms, ownership, type of house, etc. and
(c) population characteristics by block: age, religion, marital status, education, occupation, etc. There are 245 variables available in these three categories. Blocks without population are excluded from the analysis.

**Flood Model**
In this study, we will use the inundation results obtained by Somos-Valenzuela et al. (2016) that considers that an avalanche of rocks and ice could potentially fall into Palcacocha Lake and produce a chain of events that would lead to flooding in Huaraz. From all the scenarios analysed, in this study, we will use the scenario in which an avalanche of 3 million cubic meters falls into Palcacocha Lake producing a wave that overtops the moraine dike and inundates Huaraz. In Figure 3 (0 m Lowering), we
show the physical hazard map for that scenario with no mitigation.
<p style="text-align:center">Insert Figure 3</p>

**2.2.2 Evacuation Model**
To estimate the percentage of people that evacuate we use the LIFESim model as a base framework. The Army Corps of Engineering incorporated this model into the HEC-Fia model (Lehman &
Needham, 2012; USACE, 2012) to evaluate the evacuation during flood events. LIFESim has three modules: 1) Warning and Evacuation, 2) Loss of Shelter, including prediction of building performance, and 3) Loss of Life calculation.
To estimate the number of people that can perish during a flood event we need to divide the calculation into two main processes. First, we need to estimate the number of people at risk (Npar) that are not able
to escape before a flood arrives, or what it is known as the number of people exposed to risk ($N_{exp}$). Second, we need to calculate the percentage from $N_{exp}$ that can survive once they are in the inundation zone. This paper deals with the first process, the calculation of $N_{exp}$ by including social vulnerability.

---

[1] CASEN is a tool to describe and analyze the socio-economic situation of Chilean families, including housing, education, and labour characteristics. This is a cross-sectorial survey, whose periodicity yields a time based picture of the evolution of individual/household welfare (Contreras 2001).

Explaining why people evacuate faster, slower, or not at all is a process with many layers that is not easy to quantify. In the literature it is possible to recognize marked processes that can be generalized in Equation 1. First, we need to know the fraction of people that can escape (FE), for which we need to know how much time people have to escape (TE) and how feasible it is that in TE people can reach a

safe area. For example, in a sudden dam breach, the maximum TE is the time that a flood has to travel from the dam to the area of interest (Graham, 2009; S. N. Jonkman et al., 2008; McClelland & Bowles, 2002). Then we have the fraction of people that can find shelter (FS) within the inundated area and finally the number of people that can be rescued (NRES)

$$N_{EXP} = (1 - FE) \cdot (1 - FS) \cdot (NPAR) - NRES \qquad (1)$$

Since we are interested in the impact of social vulnerability in the evacuation process, we reduce Equation 1 to Equation 2

$$N_{EXP} = (1 - FE) \cdot (NPAR) \qquad (2)$$

The model LIFESim provides a methodology for how to calculate FE (Aboelata & Bowles, 2005). We use LIFESim to illustrate how to apply our findings, but the accuracy of the methodology is beyond the

scope of this paper and needs further analysis. To calculate the proportion of people that escape we consider three processes: warning, mobilization, and evacuation-transportation.

**Warning**
Time is a key component of the evacuation process; therefore, an efficient EWS is crucial to saving lives. However, understanding that there is an imminent threat is not a direct process. Equation 3 from

Rogers and Sorensen (1991) is used to estimate the proportion of people that understand the alarm when they hear it or learn from others' behavior that there is an imminent hazard and they need to evacuate.

$$\frac{dn}{dt} = k \cdot (a1 \cdot a1f \cdot (N - n)) + (1 - k) \cdot (a2n \cdot (N - n)) \qquad (3)$$

Where:

$\frac{dn}{dt} =$  is the proportion of people that understand that there is imminent hazard
$k$ = percentage of people alert as a function of the broadcast system (Rogers & Sorensen, 1991)
$(1-k)$ = proportion of people left to be warned (Rogers & Sorensen, 1991)
$a_1$=effectiveness of the warning system (Table 1 from (Rogers & Sorensen, 1991))
$a_1f$ = adjustment factor by location and activity (Table 2 from (Rogers & Sorensen, 1991))

$a_2$= effectiveness of the contagion warning process (Table 1 from (Rogers & Sorensen, 1991))
$N$ = fraction that the system is designed to warned in the first 30 minutes after issuance of the warning, also referred to in Table 1 from (Rogers & Sorensen, 1991), as the 30-min limit, and $n$ = proportion of people warned.

**Mobilization Process**
After people understand that there is a treat, they start to evacuate to a safe zone. Figure 35 from Aboelata & Bowles (2005) defines mobilization curves, below we show the "improved" curves from the cited reference.
HEC-Fia, which applies a version of LIFESim, includes the activities in which people are involved at

the moment of a flood. To understand the impacts of engaging in daily activities on the evacuation, we combined the warning penetration (using sirens and tone alert radios) and the mobilization process,

including the uncertainty bounds for both processes, with a Monte Carlo simulation with 1000 samples shows that the activity, as it is described in LIFESim, that people are doing when the alarm is released does not affect the penetration of the warning.

Although the emphasis of this work is to include Social Vulnerability, it is pertinent to show a current methodology that is adapted by the U.S. Army Corps of Engineers to provide context on how our data fits into state of the art evacuation process assessments. In Figure 4 we demonstrated that according to the LifeSIM/HecFIA models the activity that people are doing when the alarm is released does not cause significant changes in the percentage of people mobilized. Therefore, we will not include activities in our calculations when we include Social Vulnerability. Additionally, at the moment of the

survey, we did not specify to the first responder to quantify the time that people take to understand the alarm (warning penetration) nor the time that it took them to get ready to evacuate (mobilization). Therefore, the answers from the first responders correspond to the penetration and mobilization processes aggregated, which is equivalent to Figure 4.

Insert Figure 4

**Escape**

In the example of the application of this methodology, we assumed that people would walk at a speed that ranges from 80-187 meter per minute with an average of 107 meters per minute (Aboelata and Bowles 2005). The shortest path was calculated using ArcGIS.

**2.2.3 Social Vulnerability Index**

One of the main critics of the use of indexes to quantify social vulnerability is the limited number of variables and the lack of connection and interrelationship among variables used by the indexes. To face these limitations, we construct a Social Vulnerability Index (SVI) by analysing census data using Principal Component Analysis (PCA) following the methodology developed by Cutter et al., (2003). The main objective of a PCA is to extract information from the variables and represent this information

as a set of new orthogonal variables called principal components.(Wold, Esbensen, & Geladi, 1987). The use of this technique allows for robust and consistent numbers of variables that can be analysed to estimate changes in social vulnerability over time (Cutter et al., 2003). We followed Schmidtlein et al. (2008), who list seven steps to calculate the Social Vulnerability Index (SVI). To construct a Social Vulnerability Index (SVI), we analyzed census data using Principal Component Analysis(PCA). This is

a multivariate technique "that analyzes a data table in which observations are described by several inter-correlated quantitative dependent variables"(Abdi & Williams, 2010). The main objective of a PCA is to extract information from the variables and represent this information as a set of new orthogonal variables called principal components. For example, PCA "provides an approximation of a data table, a data matrix, X, in terms of the product of two small matrices T and P', These matrices, T and P',

capture the essential data pattern of X" (Wold et al., 1987). The use of this technique allows for robust and consistent numbers of variables that can be analyzed to estimate changes of social vulnerability over time (Cutter et al., 2003).

We followed Schmidtlein et al. (2008), who list 7steps to calculate the Social Vulnerability Index (SVI): (1) Normalize all variables as percentage, per capita or density functions. For the purposes of this paper,

we normalized all variables as percentages; for example, the percentage of independent houses per block or the percentage of elderly people per block.  Then standardize all input (census) variables to z-scores $z = \frac{x-\mu}{\sigma}$ . This creates variables with mean 0 and standard deviation 1. (2) Perform the PCA with

the standardized input variables (z-scores). Select the number of components with eigenvalues greater than one. (3) Rotate the initial PCA solution. In our work we used a normal Kaiser varimax rotation for component selection. (4) Calculate the Kaiser-Meyer-Olkin measure of sampling adequacy (KMO) and Bartlett's test of sphericity. (5) Interpret the resulting components as to how they may influence (increase or decrease) social vulnerability and allocate signs to the components accordingly. (6) Combine the selected component scores into a univariate score using a predetermined weighting scheme. The factors are named based on variables with significant factor loading, usually greater than .3 or less than -.3. (7) Finally, we standardized the resulting scores to mean 0 and standard deviation 1. All the steps but step 7 are straightforward. In step 5, we must decide how we want to combine the different components. The first criterion is to use the scores from the PCA, adding them but assuming that all the components have the same contribution to the SVI (Cutter et al., 2003). The second criterion uses the scores from the PCA, but assigns different weights to the principal components according to the fraction of variability they explain (Schmidtlein et al. 2008). The third method also does not assume that each component contributes equally to social vulnerability, but in contrast to the second method, it multiplies each z-score by the factor load and then each component is multiplied by its explained variance. We use the first criterion, in other words, we gave the same weight to all components. The same was done by Chakraborty et al., (2005); Chen et al., (2013); Cutter et al., (2003); Fekete, (2009) and Zhang and You, (2014). Fekete (2011) provide a solid argument that explains the reason of using equal weighting which avoids adding assumptions that are qualitative and mostly not empirically supported, although it may sound intuitive to use the loading factor or the variance explained by the factor to combine the variables selected. Moreover, Roder et al., (2017) argue that there is no appropriate methodology for the calculation of the index.

## 3 RESULTS

### 3.1 Survey to first responders

Figure 5 shows the percentage of population that evacuate after the tsunami alarm was activated in neighborhoods with high, medium and low social vulnerability. Each box presents the 75th percentile (upper hinge), the median (center), 25th percentile (lower hinge) and the outlier values. Figure 5 indicates that neighborhoods with high social vulnerability systematically evacuate fewer people than areas with medium or low social vulnerability, for example, the first 5 minutes after the alarm is activated, the median (percentage of evacuation) for neighborhoods with high social vulnerability is the 20%, and 40% for medium and low social vulnerability. Figure 5 also shows that the differences in term of the percentage of evacuation decrease over time and eventually disappear after an hour since the alarm was activated.

Insert Figure 5

We test if the mean response time to the evacuation alarm between the three types of neighborhoods was statistically significant ($p>0.05$) using two methods: Anova (parametric method) and Kruskal-Wallis (non-parametric method). Table 1 shows that the differences are not statistically significant between neighborhoods using both methods; this could be due to the limited size of the

sample. In consequence, we decide to use the median rather than the mean as the middle point of the distribution of the mean response time.

Insert Table1

## 3.2 Case Study: Hypothetical Application Case of ReTSVI in Huaraz, Peru.
### 3.2.1 Social Vulnerability Index
Peru has a long history of mudflows generated from glacial lakes in the Cordillera Blanca. As global warming progresses and glaciers start shrinking at a higher rate,this problem is growing. In some cases, glaciers leave behind a weak moraine that holds a large amount of water that can suddenly release and generate floods (for more details see Carey, 2010; Hegglin & Huggel, 2008; Somos-Valenzuela et al., 2016).

Using the population census of Peru and PCA, we were able to identify 20 census variables grouped into six components that explained social vulnerability among all the neighbourhoods in Huaraz (Table 1). The first component explains 20% of the variance and identifies the wealth of each block measured by population with primary and college education, with health insurance, indigenous population, white collar jobs and households with five or more rooms. The groups most affected by natural disasters— the elderly, women, and people with disabilities— are grouped in the second component, which explains 9% of the variance. The third component describes variables linked with poverty such as illiteracy rates, the existence of informal settlements, and households without electricity. 8% of the variation in blocks is captured by this component. The fourth component identifies home-ownership and marital status; this factor explains 7% of the variance. The fifth component groups neighbourhoods with high population density and workers in blue collar jobs that are usually linked with low-income payment, insecure and more precarious work conditions. This component captures 7% of the variation in blocks. Finally, the sixth component identifies children (<1 years old) and population working in the manufacturing sector; this component explains 6% of the variance

Insert Table 1

As Figure 6 illustrates, most of the blocks located close to the Quilcay River exhibit a higher level of social vulnerability. Conversely, those blocks concentrated in the south of the city (away from the Quilcay River) are less vulnerable. Finally, the population who lives upriver, north of Huaraz, present a middle level of vulnerability with a combination of medium-low and low levels of social vulnerability.

Insert Figure 6

The proportion of high, medium and low vulnerability blocks within the inundation zone are 15%, 35 %, and 50% respectively.
### 3.2.2 Evacuation process
We calculated the percentage of people that could evacuate after a GLOF from Palcacocha Lake, Peru. An ideal EWS would release an alarm as soon as the hazard is detected. However, the protocols normally require checking multiple sensors in order to avoid a false positive error. This process delays the alarm's release consuming important time that could otherwise be used for the population to begin evacuating. We use two methodologies to estimate the proportion of inhabitants that can leave their household before the hazard strikes. First, we use the empirical equations described in the methodology,

where we assumed that different groups react and evacuate homogeneously (Figure 7). Second, we use the information provided by the first responders, census data and SVI to include social vulnerability in the evacuation process (Figure 8). In both cases, we estimate the percentage of people that evacuate if the alarm is sounded at 0, 20, 40, 60, 70, 80, 90 and 100 minutes after the inundation starts traveling from Palcacocha Lake toward Huaraz.

An obvious, but not less important finding is that as the alarm is delayed the population has less time to escape. The results also suggest that social vulnerability has a larger impact when the warning alarm is delayed. After 60 minutes, Figure 8 gets patchier, which indicates that the population has different rates of evacuation, even though they have a similar amount of time to respond. Also, when we use information from the first responders, the evacuation is faster than when we use empirical equations from LIFESim. The finding that evacuations were completed more rapidly with the earthquake/tsunami response data than with the LIFESim equations is due to the fact that, as long as the local population recognizes earthquake shaking as a tsunami warning cue, the shaking is an instantaneous broadcast mechanism (see Lindell et al., 2015; Wei et al., 2017). In those situations, k = 1 in Equation 3, which makes the time-consuming contagion process unnecessary.

Insert Figure 7

Insert Figure 8

**4 Discussion**

The literature indicates that social vulnerability has a large influence on how people respond to natural disasters. There is agreement that more vulnerable inhabitants not only suffer the most during a natural disaster but also are less resilient, which affects their ability to recover afterward. Social vulnerability is thought to be an important factor that needs to be included in evacuation analyses but there are no systematic frameworks to do so. This paper deals with this problem by proposing a methodology to integrate social vulnerability into the calculation of how people evacuate after an EWS is activated. We develop the *Response Time by Social Vulnerability Index* (ReTSVI) methodology, which is a three-step process to determine the percentage of people that would leave an area that could be potentially inundated. For doing this, we used the methods from the LIFESim model and replaced the evacuation curves to reflect the differences in the time response according to social vulnerability level.

The findings from the surveys are in agreement with the theory since the time that people take to respond increases as the vulnerability moves from low to high levels. An interesting result is shown in Figure 9, where we compare the aggregate survey responses with the evacuation responses categorized by social vulnerability level, finding that people at a medium level of vulnerability respond similarly to the aggregated values. Then, people with low and high vulnerability behave almost symmetrically around the average. If we extrapolate these results to areas where we just know from first responders the aggregated evacuation rate in time, we can apply the factors indicated in Figure 9 to make a first order approximation of the difference in the evacuation rate by the social vulnerability.

Insert Figure 9

It is important to keep in mind that the surveys were taken in one location where people are highly trained to deal with tsunamis, which may present limitations applying this model in other locations. Regardless, this is an important advancement in our ability to quantify a process that is normally only addressed with qualitative methodologies. Certainly, we need to collect more data to come up with more general approximations of the importance of social vulnerability in the evacuation.

On the other hand, there is a body of literature that does not find a connection between social vulnerability and evacuation process (i.e. Baker, 1991; Huang, Lindell, & Prater, 2016). However, this literature has been conducted during evacuation process due to Hurricanes, where the population is informed to evacuate their home with hours or days in advance. According to our result, although with no statistical significance, social vulnerability is only relevant during the first 30 minutes after the evacuation alarm is activated, after that, the response time is almost the same among neighborhoods from different levels of social vulnerability. In the case of floods, the literature suggests that social vulnerability is an important element to consider in order to understanding different behaviours during flooding evacuations. In particular, scholars have found that variables such as low household income, poor housing quality, children (Pelling, 1997), women, housewives, students (De Marchi, 2007), elderly, high population density and population with low level of education (Zhang & You, 2014) are key variables to consider to create a social vulnerability index linked to evacuations during disasters. On the other hand, we wanted to use a methodology that make use of census information without major intervention. Therefore, we extend the application of the findings from Fekete (2009) , even though this research was conducted disaster recovery rather than evacuation, who demostrate that "social vulnerability indices are a means for generating information about people potentially affected by disasters that are e.g. triggered by river-floods." Coincidently, the components selected by the criterion used and explained in this work are similar if not the same to what the literature review indicated. Therefore, we felt encouraged to use the 6 components to first explain the responder what we mean by high, medium, and low social vulnerability and to do the exercise of application in Huaraz.

## 5 Conclusion

This article proposes a methodology to incorporate social vulnerability into current methodologies to estimate the percentage of people that evacuate an inundation hazard zone. Previous research recognizes the relevance of social vulnerability; however, it fails to connect the physical vulnerability or the characteristics of an inundation event with social vulnerability. Consequently, we propose a three-step methodology to include social vulnerability that we call Response Time by Social Vulnerability Index (ReTSVI).

We provide an example of the application of ReTSVI where we surveyed first responders to estimate the aggregated time of response and the time of response by social vulnerability. Then we used census data to calculate the SVI and applied into the evacuation process to inundation in Huaraz that was estimated in a study by Somos-Valenzuela and colleagues (2016).

The survey shows that in the first five minutes there is the larger difference in time response between social groups. In this initial period 27% of the population living in neighbourhoods with high social

vulnerability evacuated, whereas 42% and 49% of people with medium and low vulnerability escape in the same period. This tendency smooths out after 15 minutes where the distances between the different groups get closer. We use the Principal Component Analysis to construct the SVI, six factors explain social vulnerability among all blocks in Huaraz (Perú) and 57% of the variance is captured by these components. Socioeconomic status, age, gender, marital status, labour sector, education level, home-ownership, population density, poverty, and quality of dwelling materials explain the differences in social vulnerability in Huaraz.

The results of the example of ReTSVI in Huaraz highlight the relevance of including social vulnerability in the planning process. There are distinct differences in the percentage of people evacuated in Huaraz for blocks that are close to each other, which could be explained by SVI since their exposure to the physical hazard and the distance to escape are similar. The same is true when the alarm is delayed, the longer it takes for the authorities to warn people, the larger the influence of SVI. However, we have to mention that although it seems intuitively plausible that people with different levels of social vulnerability would differ in their evacuation rates and departure times, there are no empirical data that support this assumption. Differences in evacuation rate associated to level of social vulnerability needs further study because with the current state of the art and the data collected in this study, we cannot answer this question with statistical significance.

## Acknowledgements

We would like to acknowledge the Ministry of Environment of Peru for providing the 2007 Census of Peru. We also thank Cesar Portocarrero for inspiring us to develop new methodologies to help the ones in need. Finally, we would like to thank Luis Rios-Cerda for his help applying the survey to first responders and Lindsey Carte for all her feedbacks and English review that help us to improve this study.

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

## List of Tables

| Time | Anova | Kruskal-Wallis |
|---|---|---|
| 0-5 minutes | 0.13 | 0.09 |
| 0-15 minutes | 0.44 | 0.39 |
| 0-30 minutes | 0.67 | 0.60 |
| 0-45 minutes | 0.85 | 0.87 |
| 0-60 minutes | 0.87 | 0.52 |

| Selected Census variables after PCA analysis to estimate Social Vulnerability Index (SVI) | Sign Adjustment | Components | | | | | |
|---|---|---|---|---|---|---|---|
| | | 1 | 2 | 3 | 4 | 5 | 6 |
| Household with 5 or more rooms | | .31 | | | | | |
| Population with health insurance | | .40 | | | | | |
| Population with primary education | - | -.37 | | | | | |
| Population with college education | | .43 | | | | | |
| Population with "white collar jobs" | | .40 | | | | | |
| Indigenous population | | -.35 | | | | | |
| Population with disabilities | | | .53 | | | | |
| Population older than 65 years old | + | | .53 | | | | |
| Women | | | .44 | | | | |
| Informal settlement | | | | .74 | | | |
| Household without electricity | + | | | .41 | | | |
| Illiterate population | | | | .33 | | | |
| Independent houses | | | | | .56 | | |
| House rented | - | | | | .53 | | |
| Adult population divorced | | | | | -.57 | | |
| Jobs in the commerce sector | | | | | | .61 | |
| Jobs in the construction sector | + | | | | | -.33 | |
| Number of people per square kilometer | | | | | | .52 | |
| Children less than 1 year old | | | | | | | .59 |
| Jobs in the manufacturing sector | + | | | | | | .66 |
| % of variance explained by component | | 20% | 9% | 8% | 7% | 7% | 6% |
| Cumulative explained variance | | 20% | 29% | 37% | 44% | 51% | 57% |

# List of Figures

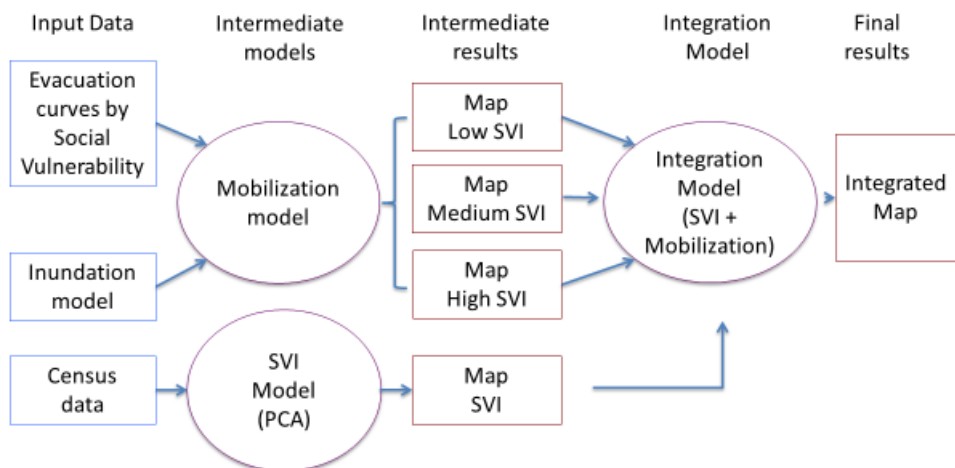

**Figure 1: ReTSVI chart**

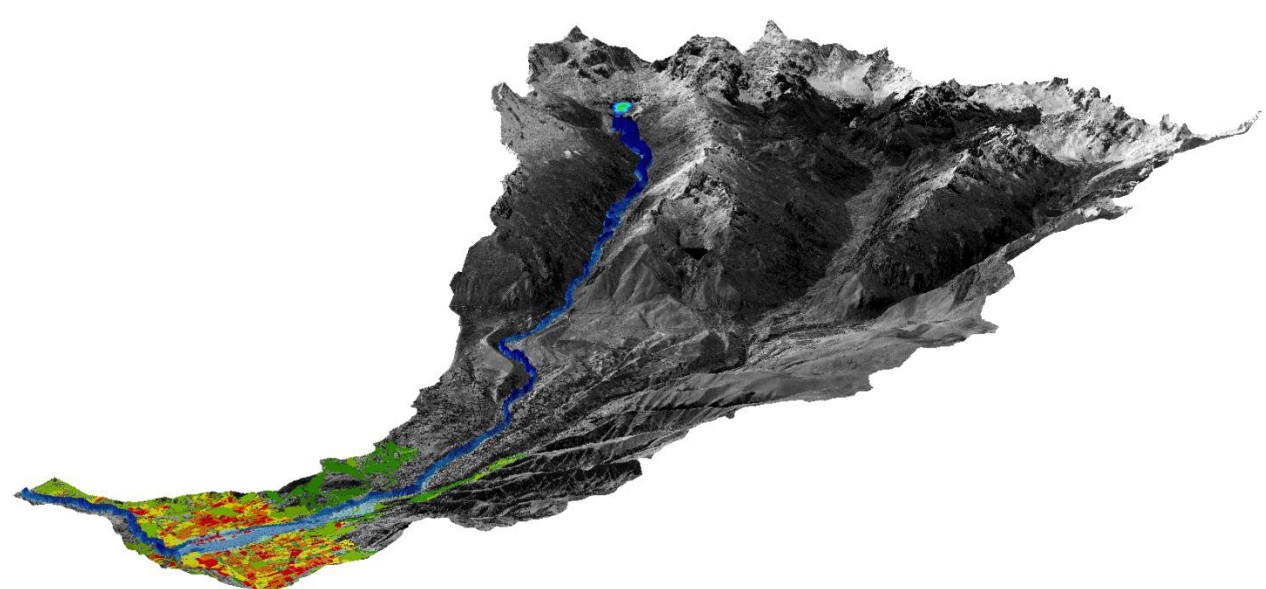

5    **Figure 2: Huaraz City in Peru at the bottom of the Cojup River. Palcacocha Lake, a potential source of a GLOF, is located at the head of the river.**

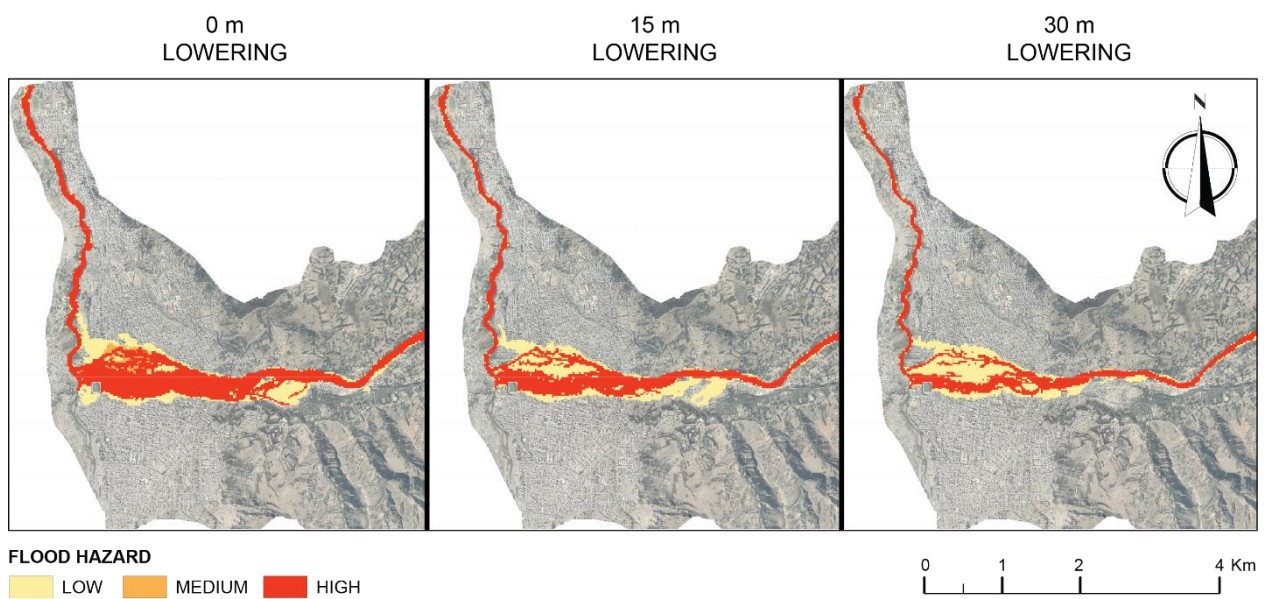

**Figure 3: This image corresponds to Figure 9 from (Somos-Valenzuela et al., 2016). Preliminary hazard map of Huaraz due to a potential GLOF originating from Lake Palcacocha with the lake at its current level (0 m lowering) and for the two mitigation scenarios (15 m lowering, and 30 m lowering).**

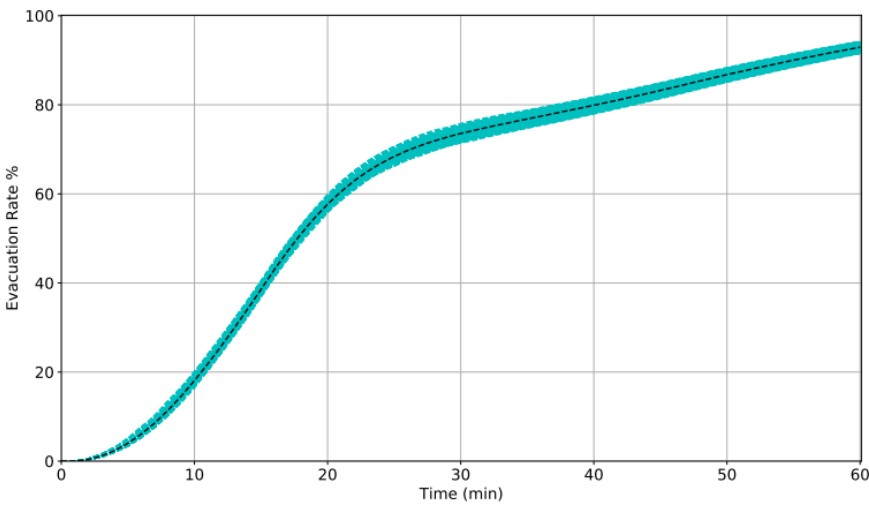

**Figure 4: Evacuation rate during the first hour calculated using 1000 samples in a Monte Carlo Simulation**

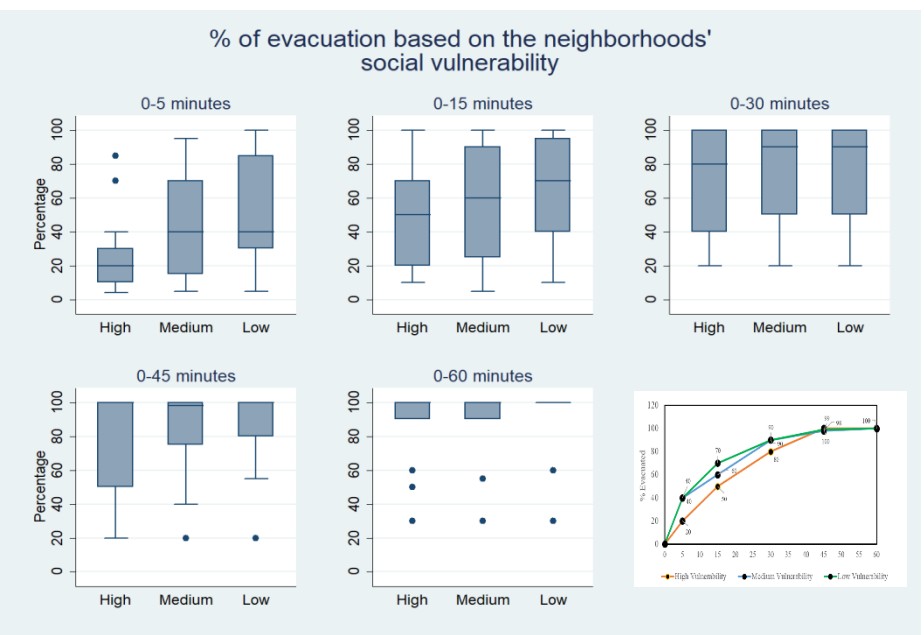

**Figure 5: First responder's results by social vulnerability group.**

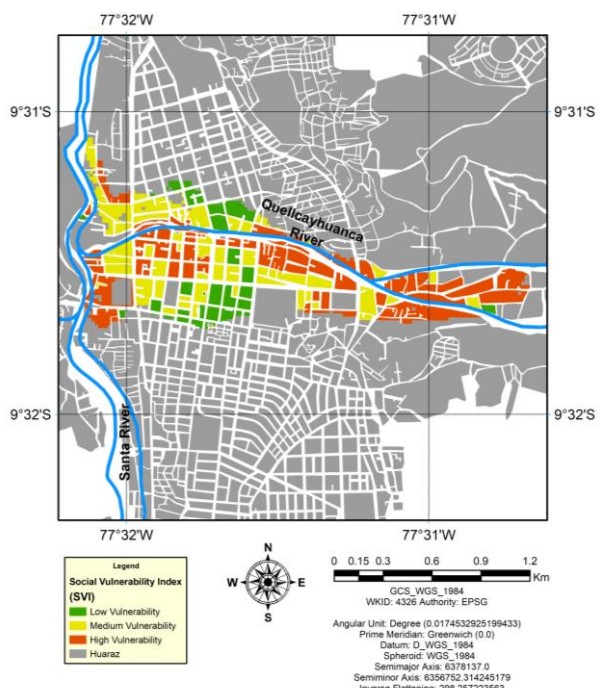

5    **Figure 6: Comparative Vulnerability of Blocks in Huaraz using Social Vulnerability Index (SVI)**

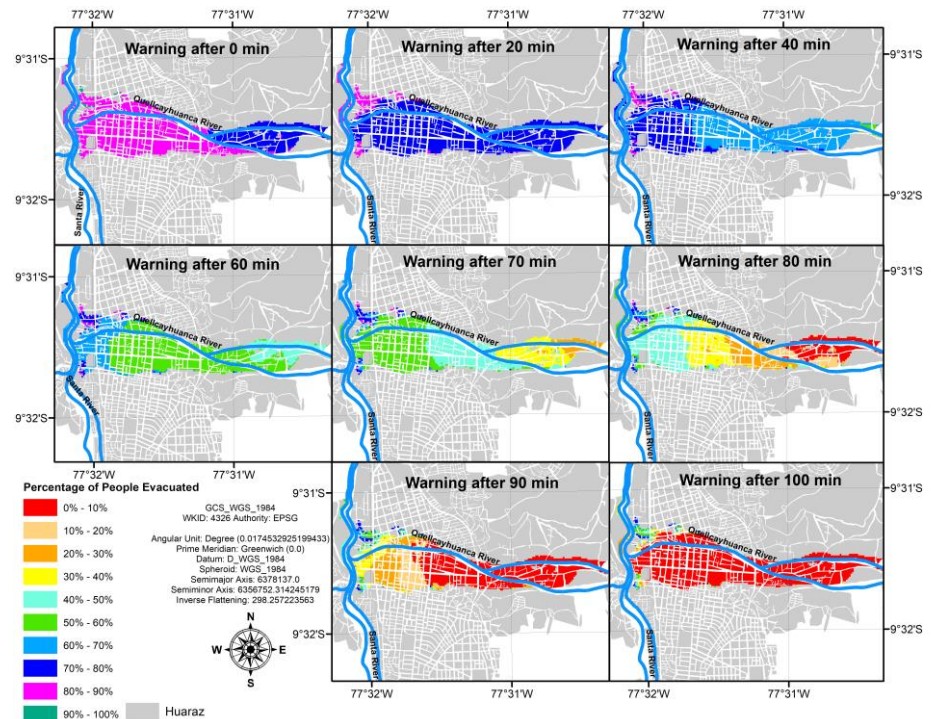

**Figure 7: Evacuation using empirical equations.**

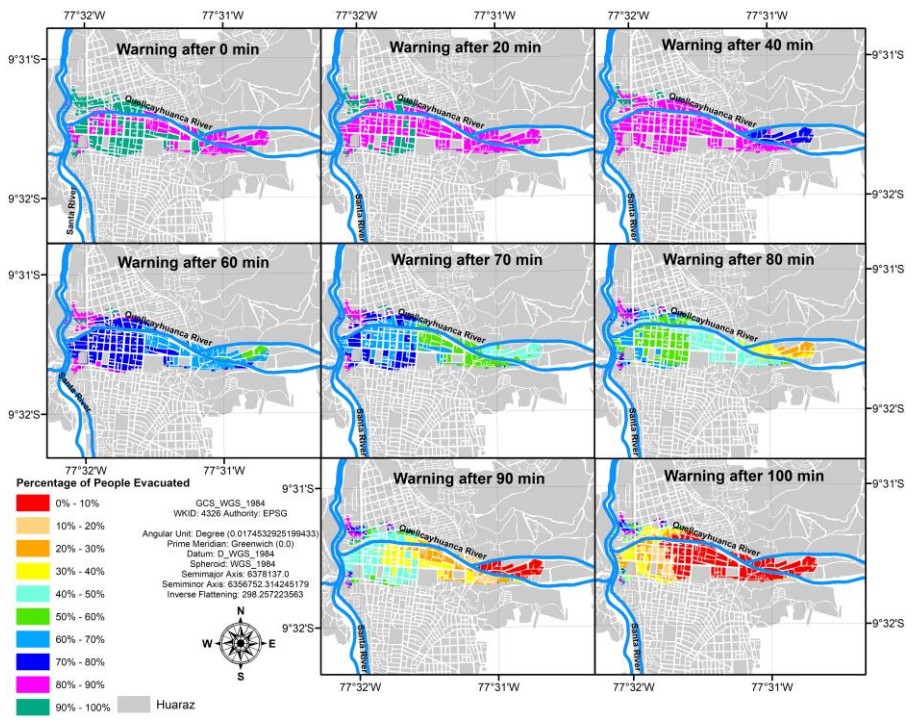

5    **Figure 8: Evacuation using Social Vulnerability Index.**

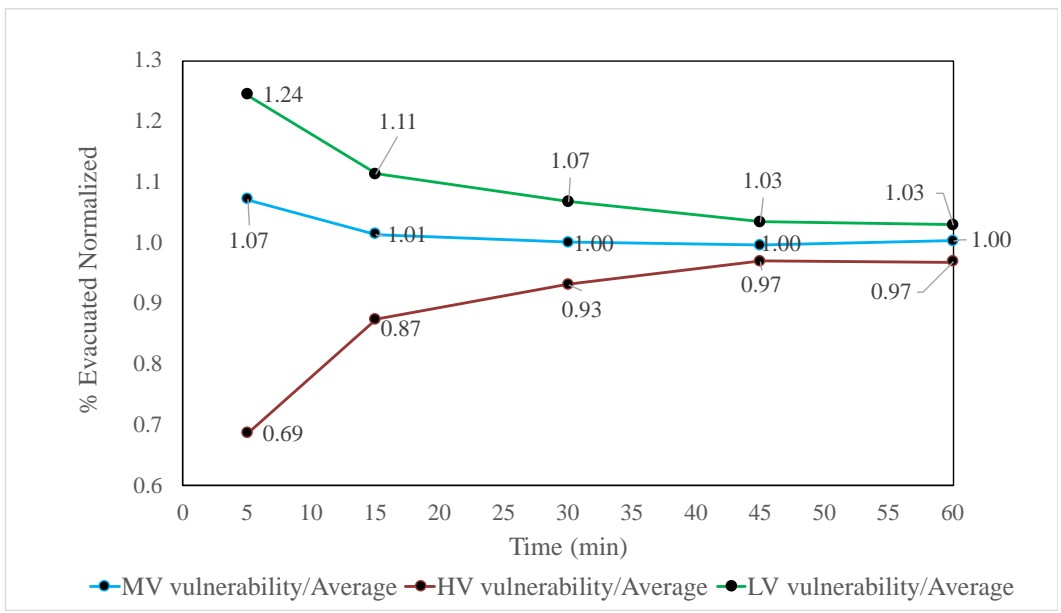

**Figure 9: People evacuated per social vulnerability level normalized by the average number of people evacuated.**