# Peer review of "Response Time to Flood Events using a Social Vulnerability Index (ReTSVI)"

_Natural Hazards and Earth System Sciences, 2017_

## Referee Comment (RC1) · Anonymous Referee #1 · 21 Dec 2017

Summary: This manuscript describes a promising method of incorporating social vulnerability into evacuation analyses. The review of the social vulnerability literature is relatively strong but the review of research on evacuation analysis is rather weak. Two very extensive reviews of research on hurricane evacuation concluded that sociodemographic variables have weak and inconsistent correlations with evacuation decisions (Baker, 1991; Huang et al., 2016) and the research on evacuation departure times is extremely sparse, even for hurricane evacuations. There is a more directly relevant literature on pedestrian evacuation for tsunamis (see the references cited below) but it does not address social vulnerability to any significant extent. In addition, there are also some unanswered questions about the reliability and validity of the evacuation departure time data reported in this study. Overall, the weak empirical foundation in the

existing literature and in this study suggests that the authors should be very cautious about any claims about the contribution that social vulnerability indicators can make in improving evacuation analysis.

Page, Line, Comment 10 L12. The description of the data from the first responders lacks specificity about the process by which the data were collected. One possibility is that each responder was asked to describe the response curve for a specific neighborhood that she or he assisted in evacuating, after which the authors classified the neighborhoods in terms of their social vulnerability. Alternatively, all responders might have been asked to generate separate curves for low, medium, or high vulnerability neighborhoods. The first procedure is much more likely than the second one to generate reliable data. The description of the data also lacks any measures of interrater agreement for the ratings of the percent evacuated at each point in time. The authors should present some measure of variability such as the standard error of the mean for each point in Figure 5. That information should be accompanied by statistical tests of the differences among the curves for low, medium, or high vulnerability neighborhoods. Given the small sample of responders, it seems quite possible that there are no statistically significant differences among the curves even at 5 minutes. If there are nonsignificant differences among social vulnerability neighborhoods at any given time point, the most appropriate estimate of percentage of evacuees at each point in time would be the median estimate. For example, Figure 5 shows that there is almost certain to be a nonsignificant difference among neighborhoods at 60 minutes. Thus, the median of the three estimates (the estimate of .89 for moderate vulnerability) would be the most appropriate statistical estimate for all three levels of social vulnerbility. If there are significant differences at some time points, then those significantly different estimates should be used. However, all time points at which there are nonsignificant differences should have the high and low vulnerability estimates replaced by the median estimate for that time point (the estimate for the moderate vulnerability group).

11 L8. If all six components were included in the SVI, what is the justification for believ-

ing that all of them are relevant to evacuation vulnerability? This issue of evacuation vulnerability (as distinct from general social vulnerability) is important because most of the Cutter et al. (2003) examples of social vulnerability in their Table 1 refer to disaster recovery rather than evacuation. There are some authors that have addressed evacuation vulnerability but, to the best of my knowledge, only Chakraborty et al. (2005) and Kusenbach et al. (2010) have examined social vulnerability in evacuation. (Cova's papers on evacuation vulnerability examine vulnerability due to evacuation route system geometry and link capacity.) Even the Chakraborty and Kusenbach studies assumed that their measures of social vulnerability would actually make a difference in evacuation rather than demonstrated it empirically. There is a broader literature on household evacuation, but the available data show no evidence that any of the sociodemographic variables measured in these studies is consistently related to evacuation (Baker, 1991; Huang et al., 2016), let alone evacuation departure time distributions. The only evacuation review to cite evidence in support of any relationships of sociodemographic variables with household evacuation only cited positive instances and ignored reports of nonsignificant correlations (Dash & Gladwin, 2007).

L11. Figure 6 does indeed show that there are many blocks of high social vulnerability located close to the river, but there are also blocks of medium and low vulnerability there as well. The authors' argument would be more persuasive if they would overlay the expected inundation zone onto the map and calculate the proportion of high, medium, and low vulnerability blocks within the inundation zone.

L27. The differences among the neighborhoods with respect to the outcomes of the evacuation model are necessarily a direct result of the presumed differences among the three evacuation rate curves. If the differences among the three curves are not significantly different from each other, then a single departure time curve should be used and the differences among the neighborhoods with respect to the outcomes of the evacuation model will vanish.

L29. The finding that evacuations were completed more rapidly with the earthquake/tsunami response data than with the LIFESim equations is due to the fact that, as long as the local population recognizes earthquake shaking as a tsunami warning cue, the shaking is an instantaneous broadcast mechanism (see Lindell et al., 2015; Wei et al., 2017). In those situations, $k = 1$ in Equation 3, which makes the time-consuming contagion process unnecessary.

12 L7 would be more accurate if restated with the following qualifications. Social vulnerability is thought to be an important factor that needs to be included in evacuation analyses but there are no systematic frameworks to do so. Moreover, although it seems intuitively plausible that people with different levels of social vulnerability would differ in their evacuation rates and departure times, there are no empirical data that support this assumption. One imitation of the available research is that Baker (1991) and Huang et al. (2016)—the two most relevant literature reviews—addressed (primarily vehicular) hurricane evacuation in the Unites States. It is unclear if these results would generalize to pedestrian evacuation in other countries.

L29. Morss et al. (2011) did not address any studies of evacuation, let alone the effects of social vulnerability on evacuation departure times, so the claim in this sentence about the comparability of the sample size is unsupported.

13 L4. This study does not "estimate the percentage of people that evacuate an inundation hazard zone" (my emphasis); it estimates the rate at which people evacuate an inundation zone.

References Baker, E.J. (1991). Hurricane evacuation behavior. International Journal of Mass Emergencies and Disasters, 9, 287-310.

Chakraborty, J., Tobin, G. A., & Montz, B. E. (2005). Population evacuation: assessing spatial variability in geophysical risk and social vulnerability to natural hazards. Natural Hazards Review, 6(1), 23-33.

Cova, T. J. (1999). GIS in emergency management. Geographical information systems, 2, 845-858.

Cova, T. J., & Church, R. L. (1997). Modelling community evacuation vulnerability using GIS. International Journal of Geographical Information Science, 11(8), 763-784.

Cova, T. J., Theobald, D. M., Norman, J. B., & Siebeneck, L. K. (2013). Mapping wildfire evacuation vulnerability in the western US: the limits of infrastructure. GeoJournal, 78(2), 273-285.

Cutter, S. L., Boruff, B. J., & Shirley, W. L. (2003). Social vulnerability to environmental hazards. Social science quarterly, 84(2), 242-261.

Dash, N. & Gladwin, H. (2007). Evacuation decision making and behavioral responses: Individual and household. Natural Hazards Review, 8, 69-77.

Fraser, S.A., Wood, N.J., Johnston, D.M., Leonard, G.S., Greening, P.D. and Rossetto, T. (2014). Variable population exposure and distributed travel speeds in least-cost tsunami evacuation modelling. Natural Hazards and Earth System Sciences, 14(11), 2975. http://www.nat-hazards-earth-syst-sci.net/14/2975/2014/nhess-14-2975-2014.html

Huang, S-K., Lindell, M.K. & Prater, C.S. (2016). Who leaves and who stays? A review and statistical meta-analysis of hurricane evacuation studies. Environment and Behavior, 48, 991-1029.

Kusenbach, M., Simms, J. L., & Tobin, G. A. (2010). Disaster vulnerability and evacuation readiness: coastal mobile home residents in Florida. Natural Hazards, 52(1), 79-95.

Lindell, M.K., Prater, C.S., Gregg, C.E., Apatu, E., Huang, S-K. & Wu, H-C. (2015). Households' immediate responses to the 2009 Samoa earthquake and tsunami. International Journal of Disaster Risk Reduction, 12, 328-340.

Wei, H-L., Wu, H-C., Lindell, M.K., Huang, S-K., Shiroshita, H., Johnston, D.M. &

Becker, J.S. (2017). Assessment of households' responses to the tsunami threat: A comparative study of Japan and New Zealand. International Journal of Disaster Risk Reduction, 25, 274-282.

Wood, N., Jones, J., Schmidtlein, M., Schelling, J. and Frazier, T. (2016). Pedestrian flow-path modeling to support tsunami evacuation and disaster relief planning in the U.S. Pacific Northwest. International Journal of Disaster Risk Reduction, 18, 41-55.

Wood, N.J. and Schmidtlein, M.C. (2012). Anisotropic path modeling to assess pedestrian evacuation potential from Cascadia-related tsunamis in the US Pacific Northwest. Natural Hazards, 62, 275–300.

Wood, N.J., Schmidtlein, M.C. and Peters, J. (2014). Changes in population evacuation potential for tsunami hazards in Seward, Alaska, since the 1964 Good Friday earthquake. Natural Hazards, 70, 1031–1053.

Wood, N., Wilson, R., Jones, J., Peters, J., MacMullan, E., Krebs, T., Shoaf, K. and Miller, K. (2017). Community disruptions and business costs for distant tsunami evacuations using maximum versus scenario based zones. Natural Hazards, 86, 619-643.

---

## Referee Comment (RC2) · Anonymous Referee #2 · 10 Jan 2018

The paper entitled "Response Time to Flood Events using a Social Vulnerability Index (ReTSVI)" seeks to explore a new method to convey the social vulnerable indicators together with evacuation response time under flood threat. Although worth of work, there is a need for significant reworking.

The introduction section is very general about the framework of social vulnerability (and sometimes only about vulnerability in general, lines 16-25, page 3) and it fails to interpret the studies in relation to floods hazard (for which a rich literature exists, e.g. Koks et al. 2015; Fekete 2009; Rufat et al. 2015; De Marchi and Scolobig 2012; Zhang and You 2014; Pelling 1997; Roder et al. 2017; De Marchi et al. 2007 among others). The paper needs extensive restructuring and in its current form fails to analyse the use of mapping social vulnerability for evacuation purposes for emergency manage-

ment plans. This is a particular application, and the authors were unable to provide a strong bibliography in support of this context. The identification of social vulnerability for effective early warning of disaster-related risks has not been adequately explained. There is no mention of the scale analysis at which mapping social vulnerability can be a usefulness tools for emergency management. Lines 1-15 page 3 is a repetition of the introduction, and lines 7-1 of the following page bring the reader a bit out of the general content of the manuscript. Moreover, the evacuation literature is structurally confused (please consider them disasters and not natural disasters that is quite overlooked) for which I suggest a more focused review and the strongest argumentation.

The objectives of the study are also not explained adequately.

The methodology part is a bit confused due to the presence of several small chapters that mix up the methods, data collection and the study area, also lacking a chronological sequence. Please organize the chapter is the simplest format to increase the readability (I suggest to start from the study area, data collection and methods at last). For the study area selection, there is a need to strongly justify the decision to study GLOF hazards in Peru providing some inundation zone maps and probability of occurrence details. The utility of having 22 interviews is not properly set. The four institutions have been not described and the questions are not well explained, as well as the type of those (quantitative, qualitative?). How could respondents define low, medium and high social vulnerability? Why are stakeholders assumed to know the average evacuation time and the percentage of the population that usually evacuates? Was it related to their personal experiences or have the data in support of it? Another critical error is made in creating the social vulnerability index. The authors used the receipt of Cutter without acknowledging properly the acronym (SoVI and not SVI as stated), the trademark and the complete receipt. Do the authors transformed the variables to be able to compare them (e.g. z-score normalization)? Do the authors made a multicollinearity analysis to prove that none of the variables was predictive of others? Which threshold for component selection (referring to Eigenvalues)? Which the adjusted directionality of

the components (Table 1)? The directionality is the most important part in the creation of the equation and thus the resulted index for each block. Also, in this regard, how factors have been weighted? (e.g. equally, Pareto rankings or with the variance each factor explained). The selection of social vulnerability indicators is only based on the work of Cutter et al. (2003) and this step is very reductive in relation to the objective of the research that is focused in evacuation rather than recovery. There is salient need to criticize construction of indicators to flood hazards looking at those variables that really would have an effect on peoples' capacity to evacuate. It will add important value to the paper and ensure an advancement in understanding social vulnerability for this specific hazard for Peru. It is not understood how the authors selected the variables (from 245 to 20). This is one of the most critical points in this part of the analysis. How the economic status affects people capacity to evacuate? How being divorced? Or renting a house? In addition, there have not been justified in accordance with the real vulnerability Peruvian people might face in this century. Why are women more vulnerable in Peru? Another issue emerges for gender. The impact of gender on social vulnerability to floods hazard is not unambiguous. As mentioned by Rufat et al., (2015) "women are also assigned more coping-capacities, greater commitment to knowledge of risk, and social relations. The case studies reveal that it is difficult to make generalizations about women's social vulnerability and that women's dependency and needs within the context of vulnerable populations might have been overemphasized. Even in developing countries with the most inequitable societies, gender alone is not predictive of social vulnerability because women's everyday living conditions vary across socioeconomic status, household structures, and geographic locations. Within this context, some studies found that gender had no impact on the social vulnerability in the face of floods at all". Some further discussion may seek to explore this factor. This is valid for all the variables. In this regard, Roder et al. 2017 address this specific problem of variables contextualization.

Regarding Result and Discussion, these chapters are very general. I would have expected a more depth analysis. Concerning the evacuation curves, are they different

statistically? Without this understanding, the related results seem not supported at all. The mapping of the social vulnerability (Figure 6) is meaningless without an understanding of the classification method used to show the three vulnerability classes (e.g. SD, Jenks Natural Breaks), in fact one could conclude that it is quite easy to play with those classes without knowing the distribution curve. Also, which is the minim, maximum and the average value of the index? Again the components have been just mentioned roughly for which is impossible to understand to their contribution to the vulnerability in the evacuation processes during a GLOF and specifically in Peru. I suggest strongly to provide a table with some basic statistics of the number of blocks in the three categories. Also, provide some spatial statistics to relate to the proximity to the river and to analyse the outcome map of social vulnerability overlapped with the flood hazard map. The discussion chapter is not adequately addressed. There is a lengthy introduction that sum up the justification of the research and the methodology undertaken and that present new results never presented before. I suggest entirely rearrange this chapter, enrich it and provide some consideration to flood management and early warning system.

I suggest improving the quality of all the figures.

All the other comments are made through the file.

[Figure]

RC2-supplement.pdf

[Figure]

**Supplement:**

[revised manuscript text omitted]

**5** **Figure 8:** Evacuation using Social Vulnerability Index.

[Figure]

[Figure]

[Figure]

[Figure]

**Figure 9: People evacuated per social vulnerability level normalized by the average number of people evacuated.**

---

## Author Comment (AC1) · 28 Feb 2018

February 28, 2018

Dear Reviewer 1

We thank you for taking the time to give this exhaustive review that had helped us to improve our document. We have taken your revision very seriously, and in the following pages, we provide answers to all the comments that you gave us, hoping very much that you feel that we have responded thoroughly.

Sincerely,

Marcelo Somos-Valenzuela
Corresponding author

Comments from reviewer 1

Summary: This manuscript describes a promising method of incorporating social vulnerability into evacuation analyses. The review of the social vulnerability literature is relatively strong but the review of research on evacuation analysis is rather weak. Two very extensive reviews of research on hurricane evacuation concluded that sociodemographic variables have weak and inconsistent correlations with evacuation decisions (Baker, 1991; Huang et al., 2016) and the research on evacuation departure times is extremely sparse, even for hurricane evacuations. There is a more directly relevant literature on pedestrian evacuation for tsunamis (see the references cited below) but it does not address social vulnerability to any significant extent. In addition, there are also some unanswered questions about the reliability and validity of the evacuation departure time data reported in this study. Overall, the weak empirical foundation in the existing literature and in this study suggests that the authors should be very cautious about any claims about the contribution that social vulnerability indicators can make in improving evacuation analysis.

Response to Summary:
We appreciate the comments from Referee 1, this methodology is intended to help filling the gap that exists in the combination of Physical and Social vulnerability which is traditionally accomplish by ranking them separately and combining them in a matrix generally of 3 by 3. We agree with most of the comments that Anonymous Referee 1 made, and we addressed them in the following pages. We understand the concern that Referee 1 has regarding the lack of empirical foundations which is also true for previous studies. In this work, what we are proposing is a methodology (ReTSVI) to combine Physical and Social Vulnerability by connecting a series of modules that represent processes that occur in an evacuation due to flood hazards; however, we understand that the outcome of this methodology is highly dependent on the definition of the evacuation rate curves. However, we argue that it has to be part of future studies to explain place to place if social vulnerability is statistically significant to describe differences in the evacuation rate. Although with no statistical significance, our results agree with the literature associated to Hurricanes since we found that social vulnerability has less impact after one hour of warning, which is the case in hurricanes where the warn can be given with days in advance. Therefore, we would restrict this work to floods that occurs in a timescale of an hour or less. What we hope this work will be useful for is to define a framework that helps to raise questions related to specific processes associated to social vulnerability that occur in an evacuation due to flood hazards, improve methodologies and integrate/test this new knowledge as modules in this framework.

- Comment 1: Page, Line, Comment 10 L12. The description of the data from the first responders lacks specificity about the process by which the data were collected. One possibility is that each responder was asked to describe the response curve for a specific neighborhood that she or he assisted in evacuating, after which the authors classified the neighborhoods in terms of their social vulnerability. Alternatively, all responders might have been asked to generate separate curves for low, medium, or high vulnerability neighborhoods. The first procedure is much more likely than the second one to generate reliable data. The description of the data also lacks any measures of interrater agreement for the ratings of the percent evacuated at each point in time. The authors should present some measure of variability such as the standard error of the mean for each point in

Figure 5. That information should be accompanied by statistical tests of the differences among the curves for low, medium, or high vulnerability neighborhoods. Given the small sample of responders, it seems quite possible that there are no statistically significant differences among the curves even at 5 minutes. If there are nonsignificant differences among social vulnerability neighborhoods at any given time point, the most appropriate estimate of percentage of evacuees at each point in time would be the median estimate. For example, Figure 5 shows that there is almost certain to be a nonsignificant difference among neighborhoods at 60 minutes. Thus, the median of the three estimates (the estimate of .89 for moderate vulnerability) would be the most appropriate statistical estimate for all three levels of social vulnerability. If there are significant differences at some time points, then those significantly different estimates should be used. However, all time points at which there are nonsignificant differences should have the high and low vulnerability estimates replaced by the median estimate for that time point (the estimate for the moderate vulnerability group).

Response to comment 1:

As the reviewer suggested, there are two possible ways to estimate recollect the data. We tried to use the first procedure (each responder was asked to describe the response curve for a specific neighborhood that she or he assisted in evacuating, after which the authors classified the neighborhoods in terms of their social vulnerability), but it is not possible to do it in Chile due to the lack of data. The most recent data available at household level comes from the census of population conducted in 2002. We checked this dataset, and one of the problems is that many of the new neighborhoods in Coquimbo built after 2002 are not present in the census data. The second option is that all responders might have been asked to generate separate curves for low, medium, or high vulnerability neighborhoods. In the case of Chile, this is the only option available. We use the National Socioeconomic Characterization Survey (CASEN)[1] from 2015, the same year that the earthquake/tsunami occurred, to calculate a social vulnerability index at the municipality level, following the same procedure identify in the section 2.2.3. This way we were able to identify the socioeconomic and demographic characteristics of the neighborhoods with high, medium and low social vulnerability in Chile. We incorporate this information in the survey, so the first responders could identify what neighborhood belongs to each category; all responders generate separate curves for low, medium, or high vulnerability neighborhoods. Table 1 shows the variables and levels that we use to define the neighborhoods' social vulnerability in Coquimbo.

Therefore, we added a new Table 1, and the former Table 1 now is 2 and the same happens to the next tables. Alsoo, we added on page 7 line 21 the following paragraph:
"We use the National Socioeconomic Characterization Survey (CASEN)[1] from 2015, the same year that the earthquake/tsunami occurred, to calculate a social vulnerability index at the municipality level, following the same procedure identified in the section 2.2.3. This way we were able to identify the socioeconomic and demographic characteristics of the neighborhoods with high, medium and low social vulnerability in Chile. We incorporate this information in the
* * *
[1] CASEN is a tool to describe and analyze the socio-economic situation of Chilean families, including housing, education, and labour characteristics. This is a cross-sectorial survey, whose periodicity yields a time based picture of the evolution of individual/household welfare (Contreras 2001).

survey, so the first responders could identify what neighborhood belongs to each category; all responders generate separate curves for low, medium, or high vulnerability neighborhoods."

Aditionally, we have modified Figure 5 and the section 3.1 and now it reads as follow:

**3.1 Survey to first responders**

Figure 5 shows the percentage of the population that evacuate after the tsunami alarm was activated in neighborhoods with high, medium and low social vulnerability. Each box presents the 75th percentile (upper hinge), the median (center), the 25th percentile (lower hinge) and the outlier values. Figure 5 indicates that neighborhoods with high social vulnerability systematically evacuate fewer people than areas with medium or low social vulnerability, for example, the first 5 minutes after the alarm is activated, the median (percentage of evacuation) for neighborhoods with high social vulnerability is the 20%, and 40% for medium and low social vulnerability. Figure 5 also shows that the differences in term of the percentage of evacuation decrease over time and eventually disappear after an hour since the alarm was activated.

[Figure]

**Figure 5: First responder's results by social vulnerability group. Bottom right figure show the median value.**

We test if the mean response time to the evacuation alarm between the three types of neighborhoods was statistically significant using two methods: Anova (parametric method) and Kruskal-Wallis (non-parametric method). Table 1 shows that the differences are not statistically significant between neighborhoods using both methods; this could be due to the limited size of the sample. In consequence, we decide to use the median rather than the mean as the middle point of the distribution of the mean response time.

Table 1: Parametric and non-parametric statistical difference test between level of social vulnerability.

| Time | Anova | Kruskal-Wallis |
|---|---|---|
| 0-5 minutes | 0.13 | 0.09 |
| 0-15 minutes | 0.44 | 0.39 |
| 0-30 minutes | 0.67 | 0.60 |
| 0-45 minutes | 0.85 | 0.87 |
| 0-60 minutes | 0.87 | 0.52 |

- Comment 2: Page 11 L8. If all six components were included in the SVI, what is the justification for believing that all of them are relevant to evacuation vulnerability? This issue of evacuation vulnerability (as distinct from general social vulnerability) is important because most of the Cutter et al. (2003) examples of social vulnerability in their Table 1 refer to disaster recovery rather than evacuation. There are some authors that have addressed evacuation vulnerability but, to the best of my knowledge, only Chakraborty et al. (2005) and Kusenbach et al. (2010) have examined social vulnerability in evacuation. (Cova's papers on evacuation vulnerability examine vulnerability due to evacuation route system geometry and link capacity.) Even the Chakraborty and Kusenbach studies assumed that their measures of social vulnerability would actually make a difference in evacuation rather than demonstrated it empirically. There is a broader literature on household evacuation, but the available data show no evidence that any of the sociodemographic variables measured in these studies is consistently related to evacuation (Baker, 1991; Huang et al., 2016), let alone evacuation departure time distributions. The only evacuation review to cite evidence in support of any relationships of sociodemographic variables with household evacuation only cited positive instances and ignored reports of nonsignificant correlations (Dash & Gladwin, 2007).

Response to comment 2: First of all, we would like to explain why we use 6 components instead of 10 or 11 or any number in between. To determine the number of components that will be part of the social vulnerability index, we selected those components with eigenvalues values greater than one, as the graph below shows. This criterion has been used by previous studies (Schmidtlein et al., 2008) and the methodology to construct the social vulnerability index was added, step by step, in the page 11 lines 21-40 and page 12 from lines 1-10

[Figure]

Figure 1 reviewer 1: Eigenvalues calculated using PCA analysis.

As the reviewer mentioned, there is a body of literature that does not find a connection between social vulnerability and evacuation process (i.e. Baker, 1991; Huang, Lindell, & Prater, 2016). However, this literature has been conducted during evacuation process due to Hurricanes, where the population is informed to evacuate their home with hours or days in advance. According to our result, although with no statistical significance, social vulnerability is only relevant during the first 30 minutes after the evacuation alarm is activated, after that, the response time is almost the same among neighborhoods from different levels of social vulnerability. In the case of floods, the literature suggests that social vulnerability is an important element to consider in order to understanding different behaviors during flooding evacuations. In particular, scholars have found that variables such as low household income, poor housing quality, children (Pelling, 1997), women, housewives, students (De Marchi, 2007), elderly, high population density and population with low level of education (Zhang and You, 2014) are key variables to consider to create a social vulnerability index linked to evacuations during disasters. On the other hand, we wanted to use a methodology that make use of census information without major intervention. Therefore, we extend the application of the findings from Fekete (2009) , even though this research was conducted disaster recovery rather than evacuation, who demostrate that "social vulnerability indices are a means for generating information about people potentially affected by disasters that are e.g. triggered by river-floods." Coincidently, the components selected by the criterion used and explained in this work are similar if not the same to what the literature review indicated. Therefore, we felt encouraged to use the 6 components to first explain the responder what we mean by high, medium, and low social vulnerability and to do the exercise of application in Huaraz.

We added this previous paragraph into the discussion section.

- Comment 3: Page 11 L11. Figure 6 does indeed show that there are many blocks of high social vulnerability located close to the river, but there are also blocks of medium and low vulnerability there as well. The authors' argument would be more persuasive if they would overlay the expected inundation zone onto the map and calculate the proportion of high, medium, and low vulnerability blocks within the inundation zone.

Response comment 3:
We agree with this comment and we have change Figure 6

[Figure]

For this new Figure 6:

[Figure]

**Figure 6: Comparative Vulnerability of Blocks in Huaraz using Social Vulnerability Index (SVI)**

And we added in the text after inserting Figure 6: "The proportion of high, medium and low vulnerability blocks within the inundation zone are 15%, 35 %, and 50% respectively."

- Comment 4: Page 11 L27. The differences among the neighborhoods with respect to the outcomes of the evacuation model are necessarily a direct result of the presumed differences among the three evacuation rate curves. If the differences among the three curves are not significantly different from each other, then a single departure time curve should be used and the differences among the neighborhoods with respect to the outcomes of the evacuation model will vanish.

Response comment 4:
The reviewer is right here. If all the curves are not significantly different from each other, at the end the result will be the same to what it is traditionally used, which is a single curve that does not discriminate the population by the social vulnerability. Therefore, this methodology is building from what it is already out there, and it proposes a framework to incorporate information on the evacuation as a function of vulnerability level when it is available.

- Comment 5: Page 11 L29. The finding that evacuations were completed more rapidly with the earth-quake/tsunami response data than with the LIFESim equations is due to the fact that, as long as the local population recognizes earthquake shaking as a tsunami warning cue, the shaking is an instantaneous broadcast mechanism (see Lindell et al.,

2015; Wei et al., 2017). In those situations, k = 1 in Equation 3, which makes the time-consuming contagion process unnecessary.

Response comment 5:

We have included the two references and the paragraph suggested in our text and from Page 11, Line 30 after the dot it reads "The finding that evacuations were completed more rapidly with the earthquake/tsunami response data than with the LIFESim equations is due to the fact that, as long as the local population recognizes earthquake shaking as a tsunami warning cue, the shaking is an instantaneous broadcast mechanism (see Lindell et al., 2015; Wei et al., 2017). In those situations, k = 1 in Equation 3, which makes the time-consuming contagion process unnecessary."

- Comment 6: Page 12 L7 would be more accurate if restated with the following qualifications. Social vulnerability is thought to be an important factor that needs to be included in evacuation analyses but there are no systematic frameworks to do so. Moreover, although it seems intuitively plausible that people with different levels of social vulnerability would differ in their evacuation rates and departure times, there are no empirical data that support this assumption. One imitation of the available research is that Baker (1991) and Huang et al. (2016) ăˇAˇ Tthe two most relevant literature reviewsăˇAˇTaddressed (primarily vehicular) hurricane evacuation in the Unites States. It is unclear if these results would generalize to pedestrian evacuation in other countries.

Response comment 6:

We modified accordingly to the reviewer suggestion on page 12, from line 7-10

- Comment 7: Page 12 L29. Morss et al. (2011) did not address any studies of evacuation, let alone the effects of social vulnerability on evacuation departure times, so the claim in this sentence about the comparability of the sample size is unsupported.

Response comment 7:

We apologize for this mistake, and we delete the sentence and reference.

- Comment 8: Page 13 L4. This study does not "estimate the percentage of people that evacuate an inundation hazard zone" (my emphasis); it estimates the rate at which people evacuate an inundation zone.

Response comment 8:

The first part of the methodology proposed is to estimate the rate at which people evacuate an inundation hazard zone for three level of social vulnerability (Figure 1). However, when it is combined with the arrival time of the flood and the evacuation mechanism (in our case walking), it is possible to calculate the percentage that departs and reach a safe area before the flood arrives. Therefore, the result of this methodology is the percentage of people that evacuate an inundation hazard zone. Figure 7 and 8 show this, the only difference between the different frames in each figure is that we highlight the effect of delaying the warning, but all of them show the percentage of people evacuated in each scenario according to the assumptions and simplifications we made.

References

Baker, E.J. (1991). Hurricane evacuation behavior. International Journal of Mass Emergencies and Disasters, 9, 287-310.

Chakraborty, J., Tobin, G. A., & Montz, B. E. (2005). Population evacuation: assessing spatial variability in geophysical risk and social vulnerability to natural hazards. Natural Hazards Review, 6(1), 23-33.

Cova, T. J. (1999). GIS in emergency management. Geographical information systems, 2, 845-858.

Cova, T. J., & Church, R. L. (1997). Modelling community evacuation vulnerability using GIS. International Journal of Geographical Information Science, 11(8), 763-784.

Cova, T. J., Theobald, D. M., Norman, J. B., & Siebeneck, L. K. (2013). Mapping wildfire evacuation vulnerability in the western US: the limits of infrastructure. GeoJournal, 78(2), 273-285.

Cutter, S. L., Boruff, B. J., & Shirley, W. L. (2003). Social vulnerability to environmental hazards. Social science quarterly, 84(2), 242-261.

Dash, N. & Gladwin, H. (2007). Evacuation decision making and behavioral responses: Individual and household. Natural Hazards Review, 8, 69-77.

Fraser, S.A., Wood, N.J., Johnston, D.M., Leonard, G.S., Greening, P.D. and Rossetto, T. (2014). Variable population exposure and distributed travel speeds in least-cost tsunami evacuation modelling. Natural Hazards and Earth System Sciences, 14(11), 2975. http://www.nat-hazards-earth-syst-sci.net/14/2975/2014/nhess-14-2975-2014.html

Huang, S-K., Lindell, M.K. & Prater, C.S. (2016). Who leaves and who stays? A review and statistical meta-analysis of hurricane evacuation studies. Environment and Behavior, 48, 991-1029.

Kusenbach, M., Simms, J. L., & Tobin, G. A. (2010). Disaster vulnerability and evacuation readiness: coastal mobile home residents in Florida. Natural Hazards, 52(1), 79-95.

Lindell, M.K., Prater, C.S., Gregg, C.E., Apatu, E., Huang, S-K. & Wu, H-C. (2015). Households' immediate responses to the 2009 Samoa earthquake and tsunami. International Journal of Disaster Risk Reduction, 12, 328-340.

Wei, H-L., Wu, H-C., Lindell, M.K., Huang, S-K., Shiroshita, H., Johnston, D.M. &Becker, J.S. (2017). Assessment of households' responses to the tsunami threat: A comparative study of Japan and New Zealand. International Journal of Disaster Risk Reduction, 25, 274-282.

Wood, N., Jones, J., Schmidtlein, M., Schelling, J. and Frazier, T. (2016). Pedestrian flow-path modeling to support tsunami evacuation and disaster relief planning in the U.S. Pacific Northwest. International Journal of Disaster Risk Reduction, 18, 41-55.

Wood, N.J. and Schmidtlein, M.C. (2012). Anisotropic path modeling to assess pedestrian evacuation potential from Cascadia-related tsunamis in the US Pacific Northwest. Natural Hazards, 62, 275–300.

Wood, N.J., Schmidtlein, M.C. and Peters, J. (2014). Changes in population evacuation potential for tsunami hazards in Seward, Alaska, since the 1964 Good Friday earthquake. Natural Hazards, 70, 1031–1053.

Wood, N., Wilson, R., Jones, J., Peters, J., MacMullan, E., Krebs, T., Shoaf, K. and Miller, K. (2017). Community disruptions and business costs for distant tsunami evacuations using maximum versus scenario based zones. Natural Hazards, 86, 619-643.

---

## Author Comment (AC2) · 28 Feb 2018

February 28, 2018

Dear Reviewer 2

We thank you for taking the time to give this exhaustive review that had helped us to improve our document. We have taken your revision very seriously, and in the following pages, we provide answers to all the comments that you gave us, hoping very much that you feel that we have responded thoroughly.

Sincerely,

Marcelo Somos-Valenzuela
Corresponding author

Comments from reviewer 2

The paper entitled "Response Time to Flood Events using a Social Vulnerability Index (ReTSVI)" seeks to explore a new method to convey the social vulnerable indicators together with evacuation response time under flood threat. Although worth of work, there is a need for significant reworking.

- Comment 1:

The introduction section is very general about the framework of social vulnerability (and sometimes only about vulnerability in general, lines 16-25, page 3) and it fails to interpret the studies in relation to floods hazard (for which a rich literature exists, e.g. Koks et al. 2015; Fekete 2009; Rufat et al. 2015; De Marchi and Scolobig 2012; Zhang and You 2014; Pelling 1997; Roder et al. 2017; De Marchi et al. 2007 among others).

Response to comment 1: We appreciate this suggestion and we modify this text from page 2 line 29-34 which originally read:

"To address this problem, some scholars have mapped physical and social vulnerability to visualize how they overlap. They have also combined them using arithmetic operations such as multiplication or addition of social and physical vulnerability indexes to create a unique indicator that considers both vulnerabilities (Cutter & Emrich, 2006; Hegglin & Huggel, 2008)".

To this: "To address this problem, some scholars have mapped physical and social vulnerability to visualize how they overlap. They have also combined them using arithmetic operations such as multiplication or addition of social and physical vulnerability indexes to create a unique indicator that considers both vulnerabilities to study evacuation (Chakraborty, Tobin, & Montz, 2005)) or recovery process after hazards occur (Cutter & Emrich, 2006; Hegglin & Huggel, 2008)"

Additionally, after page 3 line 21 after the dot to line 38 we added the following paragraph: "Models of social vulnerability, in this area, have been used to explain the capability of communities to face and recover from disasters (Chakraborty et al., 2005).

Scholars have tried to understand whether socioeconomic and demographic characteristics of the population are relevant to understand why neighborhoods or communities respond differently during an evacuation, why some people evacuate, and others do not evacuate during disasters. The evidence about evacuations during hurricanes shows mixed results. Huang, Lindell, & Prater (2016) analyzed 49 studies linked to evacuations to hurricane warnings conducted since 1991 and concluded that demographics variables have a minor or inconsistent impact on household evacuations. In contrast, others studies show that social vulnerability is a key factor to take into account during emergency management and evacuation planning (Bateman and Edwards, 2002; Chakraborty et al., 2005; Dash and

Gladwin, 2007; Kusenbach et al., 2010). In the case of floods, studies suggest that social vulnerability is an important element to consider in order understanding different behaviors during flooding evacuations. In particular, scholars have found that variables such as low household income, poor housing quality, children (Pelling, 1997), women, housewives, students (De Marchi, 2007), elderly, high population density and population with low level of education (Zhang and You, 2014) are key variables to consider to create a social vulnerability index linked to evacuations during disasters."

- Comment 2: The paper needs extensive restructuring and in its current form fails to analyze the use of mapping social vulnerability for evacuation purposes for emergency management plans. This is a particular application, and the authors were unable to provide a strong bibliography in support of this context.

Response to comment 2: We appreciate this comment and literature suggested by reviewer 1 and 2. We have modified the introduction to narrow our review toward this particular application (see our response above). Regarding restructuring the paper, we have had several native english speaker readers that have helped us to shape this document. Therefore, we feel that, unless the editor thinks otherwise, the paper flow is adequate and it can be easily followed and understood.

- Comment 3: The identification of social vulnerability for effective early warning of disaster-related risks has not been adequately explained. There is no mention of the scale analysis at which mapping social vulnerability can be a usefulness tools for emergency management.

In this work, we do not question if the social vulnerability can be useful for emergency management because it is a normal practice that is widely used. What we identified is that traditionally this process is qualitative where social vulnerability is used to aggregate the population into high, medium and low level of vulnerability. Therefore, we are proposing a methodology to push this use of social vulnerability into a quantifiable unit by including it in the evacuation process. The statistical significance is still an issue that for the number of the first responders we used and we can not solve it in this work and we provide a review for that as well. Please see modified section 3.1 and Table 1.

- Comment 4:

Lines 1-15 page 3 is a repetition of the introduction, and lines 7-1 of the following page bring the reader a bit out of the general content of the manuscript.

Response to comment 4: To avoid redundancy, we deleted page 2 lines 32-35

- Comment 5:

Moreover, the evacuation literature is structurally confused (please consider them disasters and not natural disasters that is quite overlooked) for which I suggest a more focused review and the strongest argumentation.

Response to comment 5: We used the term "natural disaster" instead of "disasters" because the nature of the problems analyzed and the spirit of this work are associated with the environment. These disasters may be trigger by human actions, but they are understood as natural events in the literature. Additionally, we extended our literature review to address the evacuation associated with natural disasters. On the other hand, if the editor suggests that we use the word "disaster" instead of "natural disaster" we will change it in the document.

- Comment 6:

The objectives of the study are also not explained adequately.

Response to comment 6: In section 2.1 Conceptual model of ReTSVI, we implicitly explain the objective of the study "The Response Time by Social Vulnerability Index (ReTSVI) methodology allows for the inclusion of social vulnerability into the traditional evacuation/mobilization models. Figure 1 is a chart of ReTSVI, we use three types of input data, which are: 1) the evacuation curves, one for each level of vulnerability (high, medium and low vulnerability); 2) a model that describes the physical hazard that the population may be exposed to, for example, the time that a flood takes to reach a populated area; and 3) demographic information such as a census data that allows us to categorize the population into different levels of social vulnerability. Then we have two intermediate models. The first one corresponds to the mobilization model that combines the evacuation curves and the inundation model. The result of this step are three maps (one for each level of vulnerability) of the percentage of people that evacuate before the flood strikes a place. The second intermediate model is the calculation of the social vulnerability index (SVI) using the census data, which produces a map of the city in which we can classify each block by social vulnerability. Finally, we combined the results (Integration Model Figure 1) from the mobilization model and the SVI calculations to generate a map with the percentage of people that can evacuate, which considers their social vulnerability level."

In order to further attend this comment, we modified the paragraph indicate above in the main document and now it reads like this: "The objective of this work is to propose a conceptual model 'The Response Time by Social Vulnerability Index (ReTSVI)' methodology that allows for the inclusion of social vulnerability into the traditional evacuation/mobilization models and it moves away from traditional methods that combined social vulnerability and hazard magnitude by ranking in a matrix system that results in

qualitative assessment. Figure 1 is a chart of ReTSVI, we use three types of input data, which are: 1) the evacuation curves, one for each level of vulnerability (high, medium and low vulnerability); 2) a model that describes the physical hazard that the population may be exposed to, for example, the time that a flood takes to reach a populated area; and 3) demographic information such as a census data that allows us to categorize the population into different levels of social vulnerability. Then we have two intermediate models. The first one corresponds to the mobilization model that combines the evacuation curves and the inundation model. The results of this step are three maps (one for each level of vulnerability) of the percentage of people that evacuate before the flood strikes a place. The second intermediate model is the calculation of the social vulnerability index (SVI) using the census data, which produces a map of the city in which we can classify each block by social vulnerability. Finally, we combined the results (Integration Model Figure 1) from the mobilization model and the SVI calculations to generate a map with the percentage of people that can evacuate, which considers their social vulnerability level."

- Comment 7:

The methodology part is a bit confused due to the presence of several small chapters that mix up the methods, data collection and the study area, also lacking a chronological sequence. Please organize the chapter is the simplest format to increase the readability (I suggest to start from the study area, data collection and methods at last).

Response to comment 7: We adressed this question above "Regarding restructuring the paper, we have had several native English speaker readers that have helped us to shape this document. Therefore, we feel that, unless the editor thinks otherwise, the paper flow is adequate and it can be easily followed and understood". Aditionally, we feel like Reviewer 2 suggest that the area of study and Huaraz is the center of this work; however, we used this place as an example of application of the methodology proposed ReSTVI. Therefore, the importance of the "area of study" or "the case study" is secondary and it needs to go after we explain the methodology not to confuse the readers.

- Comment 8:

For the study area selection, there is a need to strongly justify the decision to study GLOF hazards in Peru providing some inundation zone maps and probability of occurrence details.

Response to comment 8: The reason to use this GLOF hazards is that one of the authors did the simulation for a potential inundation in Huaraz as part of a project that was funded by United States Agency for International Development (USAID), Interamerican Development Bank (IBD) and The ministry of environment of Peru. During this work in Peru, we also wanted to evaluate the implication of installing an early warning system. Then, we realized that the population exposed to the potential hazard was completely different in terms of

social vulnerability, and we worked with the Ministry of Environment to have access to the Census data, which is not publicly available, to determine the different levels of social vulnerability and which group was going to be affected more or less. This work was published in Somos-Valenzuela (2014). During the work described, we realized that there was not a formal methodology to combine social vulnerability into the evacuation process, which is confirmed from our literature review and the literature recommended for both reviewers, we may still miss publications and examples from others part of the world though. Then we try to generate data on the evacuation rate and the differences in social vulnerability in the evacuation process in Huaraz; however, although there were a couple of evacuation drills organized by the civil defense of Peru in Huaraz, we were not allowed to access the information collected, if there were any information collected. After this, we decided to collect data after a tsunami in Coquimbo knowing that the hazard and the population are different; however, our goal is to provide a methodology and we provide an example of how the methodology should be applied.

For the second part of the question, the inundation maps were published in Somos-Valenzuela et al., (2016), we used the result of that work in this paper (Figure 5). The probability of occurrence is irrelevant for this work because we want to know the evacuation rate given the inundation scenario selected. Therefore, for the sake of this example, the probability of the inundation is 100% since it is the condition that has to happen to have the scenario presented as the application example in this paper. Additionally, calculating the probability of an inundation generated due to GLOF events is not straightforward given the nature of the hazard. There is not enough data to determine the frequency, location, and magnitude of those events. Additionally, the research frontier in GLOF is looking into the calculation of the probability of occurrence of GLOF, which is far from the scope of this paper, although, we are aware of the importance of the frequency of any hazard in a proper risk analysis which is not what we present in this work.

- Comment 9:

The utility of having 22 interviews is not properly set. The four institutions have been not described and the questions are not well explained, as well as the type of those (quantitative, qualitative?). How could respondents define low, medium and high social vulnerability? Why are stakeholders assumed to know the average evacuation time and the percentage of the population that usually evacuates? Was it related to their personal experiences or have the data in support of it? Another critical error is made in creating the social vulnerability index.

Response to comment 9:

The four institutions have been not described and the questions are not well explained, as well as the type of those (quantitative, qualitative?).

We explain in more detail who are the first responders that participated in the survey and how we recollect the data.

The original text from page 6 line 36 to page 7 line 7 reads as follow:

"Four institutions that work directly to help the population during the evacuation process participated in this study: the navy, the police, firefighters and the municipality of Coquimbo. Each institution selected at least five employees to respond to our questionnaire, these employees work directly during the emergency to help people evacuate their houses. The survey was completed with the help of a research assistant that conducted a personal interview with each participant. We asked first responders to estimate the average evacuation time and the percentage of the population that evacuates their households from 0 to 5 minutes, 0 to 15 minutes, 0 to 30 minutes, 0 to 45 minutes, 0 to 60 minutes in neighbourhoods with low, medium and high social vulnerability in Coquimbo."

We replace this text with the paragraph below:

"Four institutions that work directly to help the population during the evacuation process participated in this study: the navy, the police, firefighters and the emergency office from the municipality of Coquimbo. First, we contacted by phone with each institution to explain the purpose of the study and asked them if they agree to participate in the research, all of them agree. Then, a research assistant visited each institution and asked them to select at least five emergency experts to respond to our questionnaire. The main requirement was that the participants worked directly during the emergency to help people evacuate their houses. The research assistant conducted a personal interview with each participant.  We asked the first responders "In your opinion and based on your experience during the tsunami of 16th of September. Since the evacuation alarm was active, what is the evacuation time of population who live in areas of low/medium/high social vulnerability?" They needed to estimate the average evacuation time in neighborhoods with low, medium and high social vulnerability. Then, we asked "what is the percentage of the population that evacuate in the first X minutes? (X=5, 15, 30, 45, 60)" The first responders write down the percentage of the population that evacuates their households from 0 to 5 minutes, 0 to 15 minutes, 0 to 30 minutes, 0 to 45 minutes, 0 to 60 minutes in neighborhoods with low, medium and high social vulnerability in Coquimbo. The answers were recollected into two scales: percentages and average time (in minutes)

- Comment 10: How could respondents define low, medium and high social vulnerability?

Response to comment 10:

We use the National Socio-economic Characterization Survey (CASEN)[1] from 2015, same year that the earthquake/tsunami occurred, to calculate a social vulnerability index at the municipality level, following the same procedure identify in the section 2.2.3. This way we were able to identify the socioeconomic and demographic characteristics of the neighborhoods with high, medium and low social vulnerability. We incorporate this information in the survey, so the first responders could identify what neighborhood belongs to each category; all responders generate separate curves for low, medium, or high vulnerability neighborhoods."

- Comment 11: Why are stakeholders assumed to know the average evacuation time and the percentage of the population that usually evacuates? Was it related to their personal experiences or have the data in support of it?

Response to comment 11: Their information provide by first responders is base on their personal experience during the evacuation to the tsunami. This group of first responders participated actively and directly during the evacuation process; we asked them to estimate, based on their experience during the tsunami, what would be the percentage of evacuation, and an average time of the evacuation of the population of Coquimbo.

- Comment 12:

The authors used the receipt of Cutter without acknowledging properly the acronym (SoVI and not SVI as stated), the trademark and the complete receipt.

Response to comment 12: We use the methodology developed by Susan Cutter (2003) to construct the Social Vulnerability Index (SVI). However, we do not use the same variables to run the Principal Component Analysis because the census in Peru has different variables that the US census. Other authors, see Koks et al., (2015), Fekete (2009), also use Cutter's methodology to construct a social vulnerability index calling their indexes SVI. In consequence, we called the name SVI and not SoVI because they are created with a similar process but they are different indexes with different variables.

- Comment 13:

Do the authors transformed the variables to be able to compare them (e.g., z-score normalization)? Do the authors made a multicollinearity analysis to prove that none of the variables was predictive of others? Which threshold for component selection (referring to Eigenvalues. )?
* * *
[1] CASEN is a tool to describe and analyze the socio-economic situation of Chilean families, including housing, education, and labour characteristics. This is a cross-sectorial survey, whose periodicity yields a time based picture of the evolution of individual/household welfare (Contreras 2001).

Response to comment 13:

We did not include the methodology in the original text because we considered that the citation was enough. However, we are glad to provide an extensive explanation of what we did. Therefore, in the document in Page 10 line 9 after the dot we included the following paraphaph:

"To construct a Social Vulnerability Index (SVI), we analyzed census data using Principal Component Analysis(PCA). PCA is a multivariate technique "that analyzes a data table in which observations are described by several inter-correlated quantitative dependent variables"(Abdi and Williams, 2010). The main objective of a PCA is to extract information from the variables in a new set of orthogonal variables called principal components. For example, PCA "provides an approximation of a data table, a data matrix, X, regarding the product of two small matrices T and P', These matrices, T, and P,' capture the essential data pattern of X" (Wold et al., 1987). The use of this technique allows for robust and consistent numbers of variables that can be analyzed to estimate changes in social vulnerability over time (Cutter et al., 2003).
First, we identify the variables that were linearly correlated using the Variance Inflation Factors (VIF), those variables with VIF higher than 10 points were excluded from the model. Then, we followed Schmidtlein et al. (2008), who list seven steps to calculate the Social Vulnerability Index (SVI): (1) Normalize all variables as a percentage, per capita or density functions. For this paper, we normalized all variables as percentages; for example, the percentage of independent houses per block or the percentage of older adults per block. Then standardize all input (census) variables to z-scores $z = \frac{x - \mu}{\sigma}$. This creates variables with mean 0 and standard deviation 1. (2) Perform the PCA with the standardized input variables (z-scores). Select the number of components based on eigenvalues greater than one. (3) Rotate the initial PCA solution. In our work we used a normal Kaiser varimax rotation for component selection. (4) Calculate the Kaiser-Meyer-Olkin measure of sampling adequacy (KMO) and Bartlett's test of sphericity. (5) Interpret the resulting components as to how they may influence (increase or decrease) social vulnerability and allocate signs to the components accordingly. (6) Combine the selected component scores into a univariate score using a predetermined weighting scheme. The factors are named based on variables with significant factor loading, usually greater than .3 or less than -.3. (7) Finally, we standardized the resulting SVI scores to mean 0 and standard deviation 1. All the steps but step 6 are straightforward. In step 6, we must decide how we want to combine the different components. The first criterion is to use the scores from the PCA, adding them but assuming that all the components have the same contribution to the SVI (Cutter et al., 2003). The second criterion uses the scores from the PCA but assigns different weights to the principal components according to the fraction of variability they explain (Schmidtlein et al. 2008). The third method also does not assume that each component contributes equally to social vulnerability, but in contrast to the second method,

it multiplies each z-score by the factor load, and then its explained variance multiplies each component (Schmidtlein et al. 2008). We use the first criterion; we gave the same weight to all components. The same was done by Chakraborty et al., (2005); Chen et al., (2013); Cutter et al., (2003); Fekete, (2009) and Zhang and You, (2014). Fekete (2012) page 1167 provide a solid argument that explains the reason of using equal weighting which avoids adding assumptions that are qualitative and mostly not empirically supported, although it may sound intuitive to use the loading factor or the variance explained by the factor to combine the variables selected. Moreover, Roder et al., (2017) argue that there is no appropriate methodology for the calculation of the index."

- Comment 14: Which the adjusted directionality of the components (Table 1)?. The directionality is the most important part in the creation of the equation and thus the resulted index for each block. Also, in this regard, how factors have been weighted? (e.g., equally, Pareto rankings or with the variance each factor explained).

Response to comment 14: For the directionality, we indicate this with the sign in front of the variable name following Table 1 from Fekete (2009). However, for the sake of clarity, we modify Table one to clarify the directionality of the component and added a new column with the sign adjustment of the components.

Original Table 1:

| Selected Census variables after PCA analysis to estimate Social Vulnerability Index (SVI) + more vulnerable – less vulnerable | Components | | | | | |
|---|---|---|---|---|---|---|
| | 1 | 2 | 3 | 4 | 5 | 6 |
| - Household with 5 or more rooms | .31 | | | | | |
| - Population with health insurance | .40 | | | | | |
| + Population with primary education | -.37 | | | | | |
| - Population with college education | .43 | | | | | |
| - Population with "white collar jobs" | .40 | | | | | |
| + Indigenous population | -.35 | | | | | |
| + Population with disabilities | | .53 | | | | |
| + Population older than 65 years old | | .53 | | | | |
| + Women | | .44 | | | | |
| + Informal settlement | | | .74 | | | |
| + Household without electricity | | | .41 | | | |
| + Illiterate population | | | .33 | | | |
| - Independent houses | | | | .56 | | |
| + House rented | | | | .53 | | |
| + Adult population divorced | | | | -.57 | | |
| + Jobs in the commerce sector | | | | | .61 | |
| + Jobs in the construction sector | | | | | -.33 | |

| | | | | | | |
|---|---|---|---|---|---|---|
| + Number of people per square kilometer | | | | | .52 | |
| + Children less than 1 year old | | | | | | .59 |
| + Jobs in the manufacturing sector | | | | | | .66 |
| % of variance explained by component | 20% | 9% | 8% | 7% | 7% | 6% |
| Cumulative explained variance | 20% | 29% | 37% | 44% | 51% | 57% |

New version of Table 1

| Selected Census variables after PCA analysis to estimate Social Vulnerability Index (SVI) | Sign Adjustment | Components | | | | | |
|---|---|---|---|---|---|---|---|
| | | 1 | 2 | 3 | 4 | 5 | 6 |
| Household with 5 or more rooms | | .31 | | | | | |
| Population with health insurance | | .40 | | | | | |
| Population with primary education | - | -.37 | | | | | |
| Population with college education | | .43 | | | | | |
| Population with "white collar jobs" | | .40 | | | | | |
| Indigenous population | | -.35 | | | | | |
| Population with disabilities | | | .53 | | | | |
| Population older than 65 years old | + | | .53 | | | | |
| Women | | | .44 | | | | |
| Informal settlement | | | | .74 | | | |
| Household without electricity | + | | | .41 | | | |
| Illiterate population | | | | .33 | | | |
| Independent houses | | | | | .56 | | |
| House rented | - | | | | .53 | | |
| Adult population divorced | | | | | -.57 | | |
| Jobs in the commerce sector | | | | | | .61 | |
| Jobs in the construction sector | + | | | | | -.33 | |
| Number of people per square kilometer | | | | | | .52 | |
| Children less than 1 year old | + | | | | | | .59 |
| Jobs in the manufacturing sector | | | | | | | .66 |
| % of variance explained by component | | 20% | 9% | 8% | 7% | 7% | 6% |
| Cumulative explained variance | | 20% | 29% | 37% | 44% | 51% | 57% |

- Comment 15:

The selection of social vulnerability indicators is only based on the work of Cutter et al. (2003) and this step is very reductive in relation to the objective of the research that is focused in evacuation rather than recovery. There is salient need to criticize construction of indicators to flood hazards looking at those variables that really would have an effect on peoples' capacity to evacuate. It will add important value to the paper and ensure an advancement in understanding social vulnerability for this specific hazard for Peru.

Response to comment 15: Reducing this study just to the work by Cutter (2003) is not accurate, which is demonstrated by the many authors cited in this paper that used social vulnerability indexes. The basic idea is to use census data to shed some light to a very complex process which is understanding social vulnerability interactions. Advancing the research in social vulnerability is by no means an objective of this paper. Therefore we believe that although this is an interesting question it is out of the scope of this work.

- Comment 16: It is not understood how the authors selected the variables (from 245 to 20). This is one of the most critical points in this part of the analysis.

Response to comment 16: The selection of the variables to construct the SVI is explained in our respond to comment 13.

- Comment 17: How the economic status affects people capacity to evacuate? How being divorced? Or renting a house?

Response to comment 17: First of all, the goal of using the methodology selected to construct a social vulnerability index is to generate an index that is driven by Census data and the selection of variables is controlled by the results of the multicollinearity and PCA analyses. The major intervention is the assignment of the contribution sign to the vulnerability, and we support this from the literature revised. According to previous work that link social vulnerability and evacuation process due to disasters, the literature shows that socioeconomic status of families (in particular income and education) (Kusenbach et al., 2010), marital status of the household head and house ownership (Pelling, 1997) affect the ability of people and communities to respond or evacuate during a disaster. In this sense, we use variables to construct our Social Vulnerability Index (SVI). The specificity of the how this variables affect the evacuation is not studied in this paper and we rely on the information provided in previous work to do this selection.

- Comment 18:

In addition, there have not been justified in accordance with the real vulnerability Peruvian people might face in this century.

Response to comment 18: Knowing the real vulnerability of Peruvian people might face in this century is a task that we do not intent to answer. We understand that this is a titanic

task that would need a specific project and expertise to be answered and we anticipate that the results of that task would be subjected to scrutiny and qualitative criticism due to the multidimensional nature of human condition and therefore social vulnerability. Therefore, we selected this general methodology, which is well accepted, to estimate social vulnerability and to provide an example of application of the methodology proposed in this study. The advantage of the methodology is that if there is a better alternative to estimate social vulnerability it can be used replacing what we have shown here. We do not intent to claim success nor authorship on the social vulnerability index, we just used a well-known and accepted methodology.

- Comment 19:

Why are women more vulnerable in Peru? Another issue emerges for gender. The impact of gender on social vulnerability to floods hazard is not unambiguous. As mentioned by Rufat et al., (2015) "women are also assigned more coping-capacities, greater commitment to knowledge of risk, and social relations. The case studies reveal that it is difficult to make generalizations about women's social vulnerability and that women's dependency and needs within the context of vulnerable populations might have been overemphasized. Even in developing countries with the most inequitable societies, gender alone is not predictive of social vulnerability because women's everyday living conditions vary across socioeconomic status, household structures, and geographic locations. Within this context, some studies found that gender had no impact on the social vulnerability in the face of floods at all". Some further discussion may seek to explore this factor. This is valid for all the variables. In this regard, Roder et al. 2017 address this specific problem of variables contextualization.

Response to comment 19: This analysis is very important because it is key to identify if the components selected contribute or not to increase social vulnerability. In our study, we select the variables using a multicollinearity test and PCA; and we assign the contribution to the index based on the literature available. In some studies women are identified as more vulnerable to hurricane evacuation than men in  Kusenbach et al., 2010.  De Marchi (2007) recognizes women and household wives as "the most vulnerable responders in therm of anticipation defined as prior awareness of flood risk, evaluation of personal preparedness, precautionary measures adopted, knowledge of warning systems and codes"

- Comment 20:

Concerning the evacuation curves, are they different statistically? Without this understanding, the related results seem not supported at all.

Response to comment 20:

We test if the mean response time to the evacuation alarm between the three types of neighborhoods was statistically significant using two methods: Anova (parametric method) and Kruskal-Wallis (non-parametric method). Table 1 shows that the differences are not statistically significant ($p>0.05$) between neighborhoods using both methods; this could be due to the limited size of the sample. In consequence, we decide to use the median rather than the mean as the middle point of the distribution of the mean response time and added Table 1 to the document.

Table 1: Parametric and non-parametric statistical difference test between level of social vulnerability.

| Time | Anova | Kruskal-Wallis |
|---|---|---|
| 0-5 minutes | 0.13 | 0.09 |
| 0-15 minutes | 0.44 | 0.39 |
| 0-30 minutes | 0.67 | 0.60 |
| 0-45 minutes | 0.85 | 0.87 |
| 0-60 minutes | 0.87 | 0.52 |

- Comment 21:

The mapping of the social vulnerability (Figure 6) is meaningless without an understanding of the classification method used to show the three vulnerability classes (e.g. SD, Jenks Natural Breaks), in fact one could conclude that it is quite easy to play with those classes without knowing the distribution curve. Also, which is the minim, maximum and the average value of the index? Again the components have been just mentioned roughly for which is impossible to understand to their contribution to the vulnerability in the evacuation processes during a GLOF and specifically in Peru. I suggest strongly to provide a table with some basic statistics of the number of blocks in the three categories. Also, provide some spatial statistics to relate to the proximity to the river and to analyze the outcome map of social vulnerability overlapped with the flood hazard map.

Response to comment 21:

For the classification, we used three quantiles as it is shown in the figure below. The maximum value is 1.365, the minimum is -1.3425, the mean is 0.03, and the standard deviation is 0.4367

[Figure]

Figure 1 comments: Social Vulnerability Index statistics calculated in ArcGIS

[Figure]

Figure 2 comments:: Social Vulnerability Index classification calculated in ArcGIS.

The proportion of high, medium and low vulnerability blocks within the inundation zone is 15%, 35 %, and 50% respectively.

- Comment 22: The discussion chapter is not adequately addressed. There is a lengthy introduction that sum up the justification of the research and the methodology undertaken and that present new results never presented before. I suggest entirely rearrange this chapter, enrich it and provide some consideration to flood management and early warning system. I suggest improving the quality of all the figures. All the other comments are made through the file.
- Regarding Result and Discussion, these chapters are very general. I would have expected a more depth analysis.

Response to comment 22:

Figure 9 in the discussion does not provide new information, instead it presents the same results than Figure 5, which is in the result section, but in a different format. We also improved the figures, adding new feature to many of them and improving the resolution.

[revised manuscript text omitted]

Extra Supplementary Comments (SC)

- SC1: Page 2 line 5, worldwide, where?

Response to comment SC1: yes, worldwide. Now page 2 line 5 reads as follow:

"For example, worldwide natural disasters caused around 3.5 trillion US dollars in damages from 1980 5 to 2011,…"

- SC2: Page 2 line 8, Preparedness of whom? communities? rescue officers? policy makers.

Response to comment SC2: Preparedness of communities. Now page 2 line 8 before the dot, it reads as follow:

"A key strategy to reduce the loss of human life during a disaster is to improve preparedness of communities"

- SC3: Page 2 line 12, repeated work "age" and add "and gender"

Response to comment SC3: Now page 2 line 12  after the dot reads as follow "Individual characteristics such as  race, age, gender,…"

- SC4: Page 5 line 3. Can you please explain how class would affect people's decision to evacuate?

Response to comment SC4: we provided an extra reference that support the statement (Kusenbach et al. 2010). Our work is not to study the mechanisms to understand why class, gender or another variable could increase or decrease vulnerability instead we based the selection in the literature available.

- SC5: Page 6 line 6 "2.1 Conceptual model of ReTSVI." Is this chapter useful at all?

Response to comment SC5:

Yes, this is probably the most important section of this paper. The reason for this is that we are proposing a methodology ReTSVI that combines a series of modules which are pieces of information such us evacuation rate curves, mobilization, inundation models and social vulnerability indexes to create an integrated map of evacuation in a given location. We also provided an application example of this, which is important but it is not as relevant as the methodology proposed.

- SC6: Page 6  line 21 "There is the need for a strong and supported justification of the study area selection."

Response to comment SC6: We already addressed this point in this document, and we copy our answer here.

"The reason to use this GLOF hazards is that one of the authors did the simulation for a potential inundation in Huaraz as part of a project that was funded by USAID, BID and The ministry of environment of Peru. During this work in Peru, we also wanted to evaluate the implication of installing an early warning system. Then, we realized that the population exposed to the potential hazard was completely different in terms of social vulnerability, and we worked with the Ministry of Environment to have access to the Census data, which is not publicly available, to determine the different levels of social vulnerability and which group was going to be affected more or less. This work was published in Somos-Valenzuela (2014). During the work described, we realized that there was not a formal methodology to combine social vulnerability into the evacuation process, which is confirmed from our literature review and the literature recommended for both reviewers, we may still miss publications and examples from others part of the world though. Then we try to generate data on the evacuation rate and the differences in social vulnerability in the evacuation process in Huaraz; however, although there were a couple of evacuation drills organized by the civil defense of Peru in Huaraz, we were not allowed to access the information collected, if there were any information collected. After this, we decided to collect data after a tsunami in Coquimbo knowing that the hazard and the population are different; however, our goal is to provide a methodology, and we provide an example of how the methodology should be applied."

- SC7: Page 6 line 22 "I suggest to define what a GLOF is."

Response to comment SC7: The definition of GLOF is provided on the same line. GLOF stands for Glacier Lakes Outburst Flood.

- SC8: Page 8 line 5-7 "To estimate the percentage of people that evacuate we use the LIFESim model as a base framework. The Army Corps of Engineering incorporated this model into the HEC-Fia model (Lehman and Needham, 2012; USACE, 2012) to evaluate how flood events affect the evacuation during flood events." This sentence sounds odd. Please revise.

Response to comment SC8:

We modify this sentence and now it reads: "To estimate the percentage of people that evacuate we use the LIFESim model as a base framework. The Army Corps of Engineering incorporated this model into the HEC-Fia model (Lehman and Needham, 2012; USACE, 2012) to evaluate the evacuation during flood events."

- SC9: Page 10 line 2-3 "One of the main critics of the use of indexes to quantify social vulnerability is the limited number of variables and the lack of connection and interrelationship among variables used by the indexes." Already stated

Response to comment SC9:

We intentionally state this again, because it is important to present the information that follows. If the editor considers that it should not be there, we can certainly modify it.

- SC10: Page 10 line 4-5:  If you'd followed the methodology of cutter 2003 you should name the index SoVI and acknowledge it properly.

Response to comment SC10: We already addressed this point in this document, and we copy our answer here.

"We use the methodology developed by Susan Cutter (2003) to construct the Social Vulnerability Index (SVI). However, we do not use the same variables to run the Principal Component Analysis because the census in Peru has different variables that the US census. Other authors, see Koks et al., (2015), also use Cutter's methodology to construct a social vulnerability index and also Koks et al., (2015) called their index SVI. In consequence, we called the name SVI and not SOVI because they are different indexes with different variables."

- SC11: Page 11  line 30-33"The explanation for this may be that we took the surveys in Chile after an earthquake struck and produced a tsunami, and the population of Chile is well trained and experienced in knowing what to do in case an alarm is sounded warning of an imminent inundation."

This is true, so why using evacuation curve for a different hazard?

Response to comment SC11:

We have two reasons to do this; the first one is that we do not know any other source of data where the evacuation curves are discriminated by social vulnerability indexes. The second reason is that we wanted to provide an example of the methodology proposed and we used this tsunami hazard with the characteristic of the population similar to Peru (or at least closer than using curves from the US or Europe) as a proxy of flood generated by a GLOF.

- SC12: Page 20  line 6-7  **Figure 1: ReTSVI chart**

 Integrated map of..? Why the three classes of the mobilization model have been named as SVI low-medium and high? There should not have anything in common with the social vulnerability outcomes and the inundation model and the evacuation curves.

Response to comment SC12:

The reason to name them like that is that to create those maps, we used the evacuation curves that correspond to the vulnerability level. For example, if the evacuation map is SVI low, it means that we assumed that all the population evacuation rate follows the curve for low social vulnerability index. Then when we have the three maps (because we decided to aggregate the population in three groups), with the result of the SVI from the Census data we determined which evacuation rate should be used in each neighbor.

- SC13: Page 21  line 4  **Figure 3: This image corresponds to Figure 9 from (Somos-Valenzuela et al., 2016). Preliminary hazard map of Huaraz due to a potential GLOF originating from Lake Palcacocha with the lake at its current level (0 m lowering) and for the two mitigation scenarios (15 m lowering, and 30 m lowering).**

What low-medium-high stand for? Any reference to return period? What's the percentage of each level of hazard?

Response to comment SC13:

As the figure indicated it corresponds to flood hazard, for more information on how that was constructed, I would suggest referring to Somos-Valenzuela et al. (2016)

- SC14: Page 23 line 5 **Figure 8: Evacuation using Social Vulnerability Index.**

Spatial reference is missing

Response to comment SC14 :

We modified Figure 5, 6 and 7. Please see below.

[revised manuscript text omitted]

---

## Referee Report (RR1)

Manuscript Title:     Response Time to Flood Events using a Social Vulnerability Index (ReTSVI)

*Summary:* This manuscript continues to have significant problems that make it unacceptable for publication. These problems could be avoided if the authors would acknowledge that they have not identified any studies that provide empirical evidence that any of their census variables, let alone their overall social vulnerability index, is related to evacuation departure time. Consequently, their analysis doesn't show anything other than how a social vulnerability index *might be used if an empirically valid social vulnerability index were available*. For example, line 8 on page 15 would be acceptable if stated "The results of the example of ReTSVI in Huaraz show how a social vulnerability index could be used in the evacuation planning process. For example, such an analysis might show that there are distinct differences in the percentage of people evacuated in Huaraz for blocks that are close to each other". In the absence of any empirical evidence that their social vulnerability index is related to evacuation departure times, the current manuscript's conclusions are completely unsupported.

| Page | Comment |
| --- | --- |
| 3 | L29. This sentence attempts to support the authors' conclusion by contrasting the conclusions of a statistical meta analysis (Huang et al., 2016) with the results from 1) one of the studies in that SMA (Bateman and Edwards, 2002), 2) a woefully inadequate narrative review that only reported confirming evidence and disregarded disconfirming evidence (Dash & Gladwin, 2007), and 3) two studies that did not in fact test whether "social vulnerability is a key factor to take into account during emergency management and evacuation planning (Chakraborty et al., 2005; Kusenbach et al., 2010). Thus, the cited studies do not support the authors' conclusion. Indeed, the authors have ignored an important aspect of the Chakraborty et al. (2005) and Kusenbach et al. (2010) studies; both of them used a small set of rationally selected indicators of evacuation vulnerability rather than this study's index of unknown relevance to evacuation that was constructed by factor analyzing an arbitrary set of census variables. |

The inadequacy of the supposed support for a vulnerability index can also be seen in the claim "women, housewives, students (De Marchi, 2007) … are key variables to consider to create a social vulnerability index linked to evacuations during disasters." The report by De Marchi and her colleagues (2007)—which should actually be cited as De Marchi, Scolobig, Delli Zotti, & Del Zotto (2007)—did conclude on p. 190 that these demographic groups were vulnerable with respect to anticipation of hazards. However, they qualified this finding in the following paragraph by noting "[a]s to the phase of resistance and coping, the most vulnerable appear to be those with a low level of community embedding and with a low trust in local authorities. *The latter finding is just the opposite of that commented above pertaining to the anticipation phase, which proves that the same group may be more vulnerable at certain points in time and less vulnerable in others*" (my emphasis). More emphatically, De Marchi and Scolobig (2012, p. 317) state "[w]e maintain that one of the main problems with the operationalisation of the concept of vulnerability through indicators (see, for example, Blaikie et al., 1994; Hewitt, 1997; Anderson, 2000; Cutter, Boruff and Shirley, 2003) lies in a certain circularity of reasoning, whereby the relation between the property to be investigated and its indicators

is not clarified adequately. For instance, what is the justification for the greater vulnerability assigned a priori to women (gender being a commonly accepted vulnerability indicator)?"

Ultimately, as skeptical as I am about the value of a social vulnerability index derived from factor analysis of census data, I do not object to the authors using such an index in this paper as long as they make it clear that they are providing an example of how such an analysis would be done if they had a measure of social vulnerability that had demonstrated validity in predicting evacuation departure times. However, I do insist that they avoid claiming support of such an index from studies that do not provide such support.

5      L8. Lines 8-14 have two problems. First, this is a list of variables rather than a sentence. Second, this passage seems to be intended to repeat the claim of support for a census variable-based social vulnerability index. As noted above, none of these studies shows a reliable predictive relationship between the listed variables and any measure of people's "capacity to anticipate, cope with, resist, and recover from the impact of a natural hazard" (Blaikie et al. 1994, 9), let alone evacuation departure times.

L16. The authors seem to be committed to a policy of citing the results of individual studies even if those studies conflict with the evidence from a statistical meta analysis (e.g., Huang et al., 2016). If they really believe that these individual studies provide more conclusive evidence than a statistical meta analysis, they should explain their reasoning.

7      L26. The context suggests that the authors conducted 22 "interviews"; the entire process of selecting and contacting prospective respondents is a single "survey".

10     L20. A "critic" is a person who criticizes something; a "critique" is the content of the criticism.

L29-37. PCA is a well-known and noncontroversial technique; it is only necessary to report the authors' choices of how they handled missing data (pairwise, listwise, mean substitution, ), what they analyzed (correlations or covariances), which factor extraction they used, how they determined the number of factors, and which factor rotation procedure they used.

11     L1. Kaiser's criterion for the number of factors (eigenvalues greater than one) tends to extract too many factors; Cattell's scree test tends to be better (e.g., Costello & Osborne, 2005; Zwick & Velicer, 1986).

L3. Bartlett's test is performed on the correlation matrix, so this step should be listed *before* the current step (2)

13     L13. Shaking is an instantaneous broadcast mechanism only if the entire population recognizes it as a warning cue. However, many people in the Lindell et al. (2015) and

Wei et al. (2017) studies were warned by the social contagion process because some people were unaware of the connection between earthquake shaking and tsunamigenesis.

L21. The literature, especially the literature cited in this study, mostly *speculates* that "social vulnerability has a large influence on how people respond to natural disasters." This is especially true for the relationship between measures of social vulnerability and evacuation departure times.

L31. The findings from the interviews (not surveys) are *not* "in agreement with the theory" because there were no statistically significant differences among the evacuation curves for low, medium, and high social vulnerability.

L38. Although there is no justification for discussing differences among the vulnerability groups with respect to their evacuation departure time curves, it is interesting to note that the aggregate curve is somewhat similar to the evacuation departure time curves reported in Lindell et al. (2015).

14    L1-25. This paragraph mostly repeats the erroneous claims that I have noted earlier. However, lines 10 -13 go beyond the authors' previous unsubstantiated claims by reporting a difference due to social vulnerability that they admit is not statistically significant. In addition, lines 23-24 claim, without substantiation, that the components they used "are similar if not the same to what the literature review indicated."

L38 through line 3 on the next page reports differences in evacuation rates between vulnerability groups even though there are no significant differences among the evacuation curves for the three social vulnerability groups.

*References*
Costello, A.B. & Osborne, J.W. (2005). Best practices in exploratory factor analysis: Four recommendations for getting the most from your analysis. *Practical Assessment, Research & Evaluation,* 10(7).
Zwick, W. R., & Velicer, W. F. (1986). Comparison of five rules for determining the number of components to retain. *Psychological Bulletin, 99*(3), 432-442.

---

## Author Response (AR2)

Agricultural and Forestry Sciences

UNIVERSIDAD DE LA FRONTERA

*Avda. Francisco Salazar 01145, Casilla 54-D, 56-45-2325000, Temuco, Chile*

August 23, 2018

Dear Editor Dr. Paolo Tarolli

Thank you for considering our paper for further steps in this revision process. In this second iteration we submitted a revised version of our paper with changes accepted, an abstract and in this file the response point by point to the comments, list with the changes to the article, and a revised version with changes that are not accepted.

With all the best,

Dr. Marcelo Somos-Valenzuela

Corresponding author

**Reviewer 1**

**Comment 1 R1:**

*Summary:* This manuscript continues to have significant problems that make it unacceptable for publication. These problems could be avoided if the authors would acknowledge that they have not identified any studies that provide empirical evidence that any of their census variables, let alone their overall social vulnerability index, is related to evacuation departure time. Consequently, their analysis doesn't show anything other than how a social vulnerability index *might be used if an empirically valid social vulnerability index were available*. For example, line 8 on page 15 would be acceptable if stated "The results of the example of ReTSVI in Huaraz show how a social vulnerability index could be used in the evacuation planning process. For example, such an analysis might show that there are distinct differences in the percentage of people evacuated in Huaraz for blocks that are close to each other". In the absence of any empirical evidence that their social vulnerability index is related to evacuation departure times, the current manuscript's conclusions are completely unsupported.

**Response to comment 1 R1**

We agree with you and we tried, unsuccessfully, to indicate this in the first revision. Therefore, we include the paragraph suggested by you in the conclusion section.

Where originally reads:

"The results of the example of ReTSVI in Huaraz highlight the relevance of including social vulnerability in the planning process."

Now it reads:

"The results of the example of ReTSVI in Huaraz show how a social vulnerability index could be used in the evacuation planning process. For example, such an analysis might show that there are distinct differences in the percentage of people evacuated in Huaraz for blocks that are close to each other".

For "the absence of any empirical evidence that their social vulnerability index is related to evacuation departure times, the current manuscript's conclusions are completely unsupported", we reviewed the document and modified in several places to avoid this claim.

**Comment 2 R1:**

P 3 L29. This sentence attempts to support the authors' conclusion by contrasting the conclusions of a statistical meta analysis (Huang et al., 2016) with the results from 1) one of the studies in that SMA (Bateman and Edwards, 2002), 2) a woefully inadequate

narrative review that only reported confirming evidence and disregarded disconfirming evidence (Dash & Gladwin, 2007), and 3) two studies that did not in fact test whether "social vulnerability is a key factor to take into account during emergency management and evacuation planning (Chakraborty et al., 2005; Kusenbach et al., 2010). Thus, the cited studies do not support the authors' conclusion. Indeed, the authors have ignored an important aspect of the Chakraborty et al. (2005) and Kusenbach et al. (2010) studies; both of them used a small set of rationally selected indicators of evacuation vulnerability rather than this study's index of unknown relevance to evacuation that was constructed by factor analyzing an arbitrary set of census variables.

The inadequacy of the supposed support for a vulnerability index can also be seen in the claim "women, housewives, students (De Marchi, 2007) … are key variables to consider to create a social vulnerability index linked to evacuations during disasters." The report by De Marchi and her colleagues (2007)—which should actually be cited as De Marchi, Scolobig, Delli Zotti, & Del Zotto (2007)—did conclude on p. 190 that these demographic groups were vulnerable with respect to anticipation of hazards. However, they qualified this finding in the following paragraph by noting "[a]s to the phase of resistance and coping, the most vulnerable appear to be those with a low level of community embedding and with a low trust in local authorities. *The latter finding is just the opposite of that commented above pertaining to the anticipation phase, which proves that the same group may be more vulnerable at certain points in time and less vulnerable in others*" (my emphasis). More emphatically, De Marchi and Scolobig (2012, p. 317) state "[w]e maintain that one of the main problems with the operationalisation of the concept of vulnerability through indicators (see, for example, Blaikie et al., 1994; Hewitt, 1997; Anderson, 2000; Cutter, Boruff and Shirley, 2003) lies in a certain circularity of reasoning, whereby the relation between the property to be investigated and its indicators is not clarified adequately. For instance, what is the justification for the greater vulnerability assigned a priori to women (gender being a commonly accepted vulnerability indicator)?"

**Response to comment 2 R1**

We modified the paragraph to consider the comments

Before it reads:

"Scholars have tried to understand whether socioeconomic and demographic characteristics of the population are relevant to understand why neighborhoods or communities respond differently during an evacuation, why some people evacuate, and others do not evacuate during disasters. The evidence about evacuations during hurricanes shows mixed results. Huang, Lindell, & Prater (2016) analyzed 49 studies linked to evacuations to hurricane warnings conducted since 1991 and concluded that demographics variables have a minor or

inconsistent impact on household evacuations. In contrast, others studies show that social vulnerability is a key factor to take into account during emergency management and evacuation planning (Bateman & Edwards, 2002; Chakraborty et al., 2005; Dash & Gladwin, 2007; Kusenbach, Simms, & Tobin, 2010). In the case of floods, studies suggest that social vulnerability is an important element to consider in order understanding different behaviors during flooding evacuations. In particular, scholars have found that variables such as low household income, poor housing quality, children (Pelling, 1997), women, housewives, students (De Marchi, 2007), elderly, high population density and population with low level of education (Zhang & You, 2014) are key variables to consider to create a social vulnerability index linked to evacuations during disasters."

Now it reads:

"Scholars have tried to understand whether socioeconomic and demographic characteristics of the population are relevant to understand why neighborhoods or communities respond differently during an evacuation, why some people evacuate, and others do not evacuate during disasters. The evidence about evacuations during hurricanes shows mixed results. Huang, Lindell, & Prater (2016) analysed 49 studies linked to evacuations to hurricane warnings conducted since 1991 and concluded that demographics variables have a minor or inconsistent impact on household evacuations. In the case of floods, we have not found studies that link demographic and socio economic variables to the evacuation process."

**Comment 3 R1:**

Ultimately, as skeptical as I am about the value of a social vulnerability index derived from factor analysis of census data, I do not object to the authors using such an index in this paper as long as they make it clear that they are providing an example of how such an analysis would be done if they had a measure of social vulnerability that had demonstrated validity in predicting evacuation departure times. However, I do insist that they avoid claiming support of such an index from studies that do not provide such support.

**Response to Comment 3 R1**: We clarify this point through the document, particularly we addressed this concern in the discussion section.

**Comment 4 R1:** P5 L8. Lines 8-14 have two problems. First, this is a list of variables rather than a sentence. Second, this passage seems to be intended to repeat the claim of support for a census variable-based social vulnerability index. As noted above, none of these studies shows a reliable predictive relationship between the listed variables and any measure of people's "capacity to anticipate, cope with, resist, and recover from the impact of a natural hazard" (Blaikie et al. 1994, 9), let alone evacuation departure times.

**Response to Comment 4 R1**:

We agree with you, this was a repetition of what was already said, and we deleted the paragraph.

**Comment 5 R1:**

P7 L16. The authors seem to be committed to a policy of citing the results of individual studies even if those studies conflict with the evidence from a statistical meta analysis (e.g., Huang et al., 2016). If they really believe that these individual studies provide more conclusive evidence than a statistical meta-analysis, they should explain their reasoning.

**Response to Comment 5 R1**: We can say that there was no a policy, instead of an involuntary slip. So, we delete the individual studies that contradict the general finding of the meta-analysis as it was pointed in the previous responses.

**Comment 6 R1:**

P7 L26. The context suggests that the authors conducted 22 "interviews"; the entire process of selecting and contacting prospective respondents is a single "survey".

**Response to Comment 6 R1**:

We change the word "interview" for "survey".

**Comment 7 R1:**

P 10 L20. A "critic" is a person who criticizes something; a "critique" is the content of the criticism.

**Response to Comment 7 R1**:

We change the word "critic" for "critique".

**Comment 8 R1:**

L29-37. PCA is a well-known and noncontroversial technique; it is only necessary to report the authors' choices of how they handled missing data (pairwise, listwise, mean substitution, ), what they analyzed (correlations or covariances), which factor extraction they used, how they determined the number of factors, and which factor rotation procedure they used.

**Response to Comment 8 R1**:

Census data from Huaraz does not have missing values. For the factor extraction, first we perform PCA, then we selected the factors with eigenvalues greater than one (we extracted 6 components) which was corroborated with a screen test that we did not include in the

paper due to the large extend of it (see next response). Finally, we used the Kaiser varimax rotation.

**Comment 9 R1:**

P11 L1. Kaiser's criterion for the number of factors (eigenvalues greater than one) tends to extract too many factors; Cattell's scree test tends to be better (e.g., Costello & Osborne, 2005; Zwick & Velicer, 1986).

**Response to Comment 9 R1**:

We used the scree test to verify whether the extraction of the components with eigenvalues greater than 1 is adequate, which we did not included it in the previous version of the manuscript. The figure below shows that the appropriate number of components should be between 5 and 6 components. Therefore, we feel confident in our original selection of 6 components using the original criteria.

[Figure]

Figure 1 responses: Scree test

**Comment 10 R1:**

L3. Bartlett's test is performed on the correlation matrix, so this step should be listed *before* the current step (2)

**Response to Comment 10 R1**:

We delete the test from step 2 and moved it to step 1 that now reads as follows:

"(1) First we perform a multicollinearity test call Variance Inflation Factor (VIF). Variables with VIF>10 were excluded. Then, we normalize all variables as percentage, per capita or density functions. For the purposes of this paper, we normalized all variables as percentages; for example, the percentage of independent houses per block or the percentage of elderly people per block. Then standardize all input (census) variables to z-scores $z =$

$\frac{x-\mu}{\sigma}$ . This creates variables with mean 0 and standard deviation 1. Finally, we use the Bartlett's test of sphericity to determine if the variables are suitable for structure detection."

**Comment 11 R1:**

P13 L13. Shaking is an instantaneous broadcast mechanism only if the entire population recognizes it as a warning cue. However, many people in the Lindell et al. (2015) and Wei et al. (2017) studies were warned by the social contagion process because some people were unaware of the connection between earthquake shaking and tsunamigenesis.

**Response to Comment 11 R1**:

Before the earthquake/tsunami in Coquimbo described in this paper, there was 8.9 earthquake in Chile were people were unaware of the connection between earthquake shaking and tsunami genesis and many of them did not evacuate and perish. There was also confusing information from the authorities. Because of that, in Coquimbo as well as most of the coast of Chile, people were more alert and readier to evacuate even when the authorities said differently.

**Comment 12 R1:**

L21. The literature, especially the literature cited in this study, mostly *speculates* that "social vulnerability has a large influence on how people respond to natural disasters." This is especially true for the relationship between measures of social vulnerability and evacuation departure times.

**Response to Comment 12 R1**:

We modified the paragraph.

Before it read:

"The literature indicates that social vulnerability has a large influence on how people respond to natural disasters. There is agreement that more vulnerable inhabitants not only suffer the most during a natural disaster but also are less resilient, which affects their ability to recover afterward. Social vulnerability is thought to be an important factor that needs to be included in evacuation analyses but there are no systematic frameworks to do so."

Now it reads:

"The literature indicates that social vulnerability has a large influence on how people are affected by natural disasters. There is agreement that more vulnerable inhabitants not only suffer the most during a natural disaster but also are less resilient, which affects their ability to recover afterward. Social vulnerability is thought to be an important factor that needs to be included in evacuation analyses but there are no systematic frameworks to do so and

there is not strong evidence that proves that demographic and socio economic variables can explain the evacuation process"

**Comment 13 R1:**

L31. The findings from the interviews (not surveys) are *not* "in agreement with the theory" because there were no statistically significant differences among the evacuation curves for low, medium, and high social vulnerability.

**Response to Comment 13 R1**:

"The findings from the interviews (not surveys)" in a previous comment you mentioned that we should use survey and not interview so we changed accordingly, and we are using survey to refer to our data collection process.

Additionally, we modified the sentence and now it reads: "The findings from the surveys look somewhat similar with the theory, even though there were no statistically significant differences among the evacuation curves for low, medium, and high social vulnerability"

**Comment 14 R1:**

L38. Although there is no justification for discussing differences among the vulnerability groups with respect to their evacuation departure time curves, it is interesting to note that the aggregate curve is somewhat similar to the evacuation departure time curves reported in Lindell et al. (2015).

**Response to Comment 14 R1**:

Yes, this is very interesting considering the distance, and socioeconomic and cultural differences that the evacuation results in both studies show that in the first 15 minutes the aggregated evacuation rate falls between 50-65%, in 30 minutes from 80-90% and after an hour is close to 100%. It would be interesting to see what explains the differences between the inland and the coastal evacuation curves in Lindell et al (2015) considering that the differences are statistically significant (p<0.05).

[Figure]

Fig. 2. Percent of repondents reporting expected time of tsunami arrival and actual time of evacuation departure, by Location.

Figure 2 responses: Left, Aggregated evacuation rate from Lindell et al. (2015). Right, Figure 5 of this paper.

As a result, both evacuation process is considerably faster than the results for dam breaks from Aboelata et al. (2003) that was incorporated in the model LifeSIM and HEC-FIA model from the corp of engineering as it was pointed out in the paper.

[Figure]

Figure 3 responses: Existing and improved mobilization distributions Aboelata et al. (2003)

Aboelata, M., Bowles, D. S., & McClelland, D. M. (2003, October). A model for estimating dam failure life loss. In Proceedings of the Australian committee on large dams risk workshop, Launceston, Tasmania, Australia

**Comment 15 R1:**

P14 L1-25. This paragraph mostly repeats the erroneous claims that I have noted earlier. However, lines 10 -13 go beyond the authors' previous unsubstantiated claims by reporting a difference due to social vulnerability that they admit is not statistically significant. In addition, lines 23-24 claim, without substantiation, that the components they used "are similar if not the same to what the literature review indicated."

**Response to Comment 15 R1**: We have rewritten the discussion section and now it reads as follows:

**"4 Discussion**

This paper proposes a methodology to integrate social vulnerability into the calculation of the people evacuation rate after an EWS is activated. We develop the *Response Time by Social Vulnerability Index* (ReTSVI) methodology, which is a three-step process to determine the percentage of people that would leave an area that could be potentially inundated.

We found that the aggregated evacuation rate curve for the 2015 tsunami in Coquimbo has similarities with the evacuation curve for the 2009 tsunami in American Samoa after a 8.1

earthquake described in Lindell et al. (2015). This similarity is notable considering the distance, and socioeconomic and cultural differences. The evacuation results in both studies show that in the first 15 minutes the aggregated evacuation rate falls between 50-70%, in 30 minutes from 80-90% and after an hour is close to 100%. These aggregated evacuation curves for tsunamis are faster than the results from Equation 1 (Figure 4), and the results from Abolaeta et al. (2003) that deal with rivers and dam break floods, suggesting that the process is understood earlier by the population. This could be due to awareness/training or to the shaking that is felt by most of the people immediately.

When we separate the results by social vulnerability, the results suggest that people with higher level of vulnerability needs more time to evacuate than people with lower level of vulnerability. However, in our results, the differences between the evacuation curves are not statistically significant. In Figure 9, where we compare the aggregate survey responses with the evacuation responses categorized by social vulnerability level, we find that people at a medium level of vulnerability respond similarly to the aggregated values. Then, people with low and high vulnerability behave almost symmetrically around the average. Which in a more general application could be used to generate boundaries for the evacuation curves.

Insert Figure 9

To overcome the limitation of the no-significance in the difference between the evacuation curves more data need to be collected.

A limitation that arises when we apply a methodology such us ReTSVI, which relies on the construction of a social vulnerability index, is that we could not find studies that relate evacuation rates with social vulnerability for inundations that take less than an hour from the triggering to the flooding. In this study we used an SVI based on PCA to select the variables as proxy; however, this index was created and validated for post event assessments. Therefore, this is a limitation that needs to be addressed before applying this framework.

Traditionally, the evacuation rate is calculated using one evacuation rate curve; therefore, ReTSVI seeks to overcomes this limitation by allowing the user to include social vulnerability. The user decides which social vulnerability index use and the evacuation curves for the levels of vulnerability. Here we provide an example using as a proxy a social vulnerability index for post disaster and evacuation curves that have not statistical significance. However, it still provides valuable information (Figure 8) of the implications of including social vulnerability that needs to be validated. For example, more vulnerable people, according to the SVI based on PCA and census data, live closer to the river where the inundation strikes earlier and harder, having less time to evacuate while at the same time they evacuate later. Additionally, social vulnerability seems to be less important as the EWS gets delayed."

**Comment 16 R1:**

L38 through line 3 on the next page reports differences in evacuation rates between vulnerability groups even though there are no significant differences among the evacuation curves for the three social vulnerability groups.

**Response to Comment 16 R1**:

We change the paragraph.

Before it read:

"The survey shows that in the first five minutes there is the larger difference in time response between social groups. In this initial period 27% of the population living in neighbourhoods with high social vulnerability evacuated, whereas 42% and 49% of people with medium and low vulnerability escape in the same period."

Now it reads:

"The survey shows, without statistical significance (Table1), that in the first five minutes there is the larger difference in time response between social groups. In this initial period 27% of the population living in neighbourhoods with high social vulnerability evacuated, whereas 42% and 49% of people with medium and low vulnerability escape in the same period."

**Reviewer 2**

The authors responded accurately to almost all of the issues raised. However, several points need to be solved already.

**Comment 1 R2:** The authors used 0.3 and -0.3 in the factor loading. Can the authors justify their choice? Without seeing the other values, this choice is difficult to explain.

**Response comment 1:** Fekete (2009) used this criteria to select the variables to create his social vulnerability index: "For the interpretation, only eigenvalues greater than one are regarded and absolute loading values below 0.30 suppressed (Nardo et al., 2005: 40, 43; Buhner, 2006: 200, 211; Bernard, 2006: 677)".

Fekete A (2009) Validation of a social vulnerability index in context to river-floods in Germany. Nat. Hazards Earth Syst. Sci. 9(2), 393–403 (doi:10.5194/nhess-9-393-2009)

**Comment 2 R2:** I understand that advancing the research in social vulnerability is not on the scope of the current paper. However, the authors are using variables to explain people vulnerability to evacuation purposes. Thus, an approximate justification of those is

necessary to address the aim of the paper. I still have some doubts about the association between types of jobs (with a difference in commerce, construction and manufacturing) with the evacuation vulnerability. The same is for renters and houses without electricity. I find instead proper the relation with strictly socio-demographic variables that are supported by the mentioned literature (Kusenbach et al., 2010, that however does not include marital status as addressed by the authors in the rebuttal). Pelling (1997) addresses the complex issue of disaster vulnerability without any mention of evacuation. He is referring to mitigation/preparedness actions instead (e.g. pg 216). Thus I think the authors should find references supporting the variables selection or strongly justify their choice. If authors are not able to do so, I might suggest to replace those variables or delete them. This issue has nothing to deal with the issue raised in the first round of revision when asking to set up a background for variables selection in Peru.

**Response comment 2:** One of the primary variables that are linked to social vulnerability is the household income (Cutter et al. 2003), but in the case of Peru, this variable is not surveyed in the census. Consequently, we used proxy variables of income such us job types, marital status, renters and house without electricity. The relationship between income distribution and the job type has been established worldwide (Galbraith, 2001) as well as in literature linked to social vulnerability (post disaster) "Some occupations, especially those involving resource extraction, may be severely impacted by a hazard event" (Cutter, 2003 page 248). In relation to renters, they are considered to be more vulnerable to disaster "People that rent do so because they are either transient or do not have the financial resources for home ownership. They often lack access to information about financial aid during recovery. In the most extreme cases, renters lack sufficient shelter options when lodging becomes uninhabitable or too costly to afford". (Cutter 2003, page 247). Regarding to "houses without electricity," we assumed that it indicates more precarious conditions and the same is true for many other variables in the Census such as houses without restroom and tab water which were disregard in the collinearity test. Finally, "adult population divorced" as proxies to household income, for example Schoeni (1995) found that "In most cases, both separated and divorced men earn more than men who are never married but less than those who are currently married".

We recalculate the SVI excluding marital status (adult population divorced) and gender, and we found differences between the former and the recalculated SVI. However, after a lengthy discussion, we do not have strong arguments to support such changes specially because the variables initially selected are easily found in the literature to influence the level of Social Vulnerability.

Cutter SL, Boruff BJ and Shirley WL (2003) Social Vulnerability to Environmental Hazards. *Soc. Sci. Queartely* **84**(2), 242–161

Galbraith, J. K., & Berner, M. (Eds.). (2001). Inequality and industrial change: a global view. Cambridge University Press.

Schoeni, R. F. (1995). Marital status and earnings in developed countries. Journal of population economics, 8(4), 351-359.

**Comment 3 R2:** Revise carefully the new statement at pg. 6 lines 20-26 since not all the mentioned literature is related to evacuation, at all.

*Response comment 3:*

We revised the literature cite and the author is correct so we modify the paragraph

It originally read:

"In the case of evacuation process during hurricanes and floods, variables such as number of housing units, mobile homes, poverty, age, people with disabilities (Chakraborty et al., 2005), education, household income, pet ownership (Kusenbach et al., 2010), household size, elderly, children (Dash & Gladwin, 2007), household quality, community organization (Pelling, 1997), communities' immaterial characteristics as energy, vigour, vitality (De Marchi, 2007; De Marchi & Scolobig, 2012), average number of people per house, population density (person/km2), illiterate population and urban population ration (Zhang & You, 2014)"

Now it reads:

"In the case of evacuation process during hurricanes and floods, variables such as number of housing units, mobile homes, poverty, age, people with disabilities (Chakraborty et al., 2005), education, household income, pet ownership (Kusenbach et al., 2010), gender, household size, elderly, children (Dash & Gladwin, 2007)"

**Comment 4 R2:** What's the p-value obtained by Bartlett's test of sphericity?

*Response comment 4:*

The p-value is close to 0 as it is shown in figure below.

```
Bartlett test of sphericity

Chi-square        =        4014.630
Degrees of freedom =            190
p-value           =           0.000
H0: variables are not intercorrelated
```

**Comment 5 R2:** Did the authors perform a multicollinearity analysis before running the PCA to be sure that none of the variables has been taken twice?

**Response comment 5:**

Yes, we did. In the step 1 when we calculate SOVI. We unfortunately did not mention that in the original document, so we appreciate your questions. We also move the sphericity test from step 2 to step 1 as it was pointed out by another reviewer.

Step one originally read:

"(1) Normalize all variables as percentage, per capita or density functions. For the purposes of this paper, we normalized all variables as percentages; for example, the percentage of independent houses per block or the percentage of elderly people per block. Then standardize all input (census) variables to z-scores $z = \frac{x-\mu}{\sigma}$. This creates variables with mean 0 and standard deviation 1."

Now it reads:

"(1) First we perform a multicollinearity test call Variance Inflation Factor (VIF). Variables with VIF>10 were excluded. Then, we normalize all variables as percentage, per capita or density functions. For the purposes of this paper, we normalized all variables as percentages; for example, the percentage of independent houses per block or the percentage of elderly people per block. Then standardize all input (census) variables to z-scores $z = \frac{x-\mu}{\sigma}$. This creates variables with mean 0 and standard deviation 1. Finally, we use the Bartlett's test of sphericity to determine if the variables are suitable for structure detection."

**Comment 6 R2:** Can you discuss more in-depth table 1?

**Response comment 6:**

We modified the document where we talk about Table 1,

Before it said:

"We test if the mean response time to the evacuation alarm between the three types of neighborhoods was statistically significant (p>0.05) using two methods: Anova (parametric method) and Kruskal-Wallis (non-parametric method). Table 1 shows that the differences are not statistically significant between neighborhoods using both methods; this could be due to the limited size of the sample. In consequence, we decide to use the median rather than the mean as the middle point of the distribution of the mean response time."

Now it says:

"We test if the mean response time to the evacuation alarm between the three types of neighborhoods was statistically significant using two methods: Anova (parametric method) and Kruskal-Wallis (non-parametric method). The table 1 shows that the p-values between the time of response and level of social vulnerability (low, medium and high) are not statistically significant. All the p-values are higher that 0.05 (alpha level), and therefore we accept the null hypothesis that the time of responses between the three groups of social vulnerability are not statistically significant (p-value >.05). This could be due to the limited size of the sample. In consequence, we decide to use the median rather than the mean as the middle point of the distribution of the mean response time."

And we also modified table 1:

Table 1: Parametric and non-parametric statistical difference test between level of social vulnerability.

| Time | Anova (p-value) | Kruskal-Wallis (p-value) |
| --- | --- | --- |
| 0-5 minutes | 0.13 | 0.09 |
| 0-15 minutes | 0.44 | 0.39 |
| 0-30 minutes | 0.67 | 0.60 |
| 0-45 minutes | 0.85 | 0.87 |
| 0-60 minutes | 0.87 | 0.52 |

**Comment 7 R2:** Can you explain why three classes quartile has been chosen in classifying the Social Vulnerability index?

**Response comment 7:** Traditionally risk studies use three level of vulnerability which are normally present as a matrix of 3 by 3 where the physical and social vulnerability are intersected. Therefore, we decided to replicate this nomenclature to make our results easier to understand since the parallel is straightforward.

**Comment 8 R2:** I suggest providing some discussion related to the role of proximity to the river and the differences in SVI results.

**Response comment 8:** We provide some discussion this in the new reworked discussion section

**Comment 9 R2:** The discussion chapter is still not adequately addressed. There is a lengthy introduction that sums up the justification of the research and the methodology undertaken. The new chapter on page 17 lines 1-20 is a repetition of the introduction. Thus the discussion is still lacking of a new light to results. Discussions should include

similarities and criticisms to findings. For all these reasons, I suggest rearranging this chapter again.

**Response comment 9:** We have rewritten this section and now it reads as follows:

**"4 Discussion**

This paper proposes a methodology to integrate social vulnerability into the calculation of the people evacuation rate after an EWS is activated. We develop the *Response Time by Social Vulnerability Index* (ReTSVI) methodology, which is a three-step process to determine the percentage of people that would leave an area that could be potentially inundated.

We found that the aggregated evacuation rate curve for the 2015 tsunami in Coquimbo has similarities with the evacuation curve for the 2009 tsunami in American Samoa after a 8.1 earthquake described in Lindell et al. (2015). This similarity is notable considering the distance, and socioeconomic and cultural differences. The evacuation results in both studies show that in the first 15 minutes the aggregated evacuation rate falls between 50-70%, in 30 minutes from 80-90% and after an hour is close to 100%. These aggregated evacuation curves for tsunamis are faster than the results from Equation 1 (Figure 4), and the results from Abolaeta et al. (2003) that deal with rivers and dam break floods, suggesting that the process is understood earlier by the population. This could be due to awareness/training or to the shaking that is felt by most of the people immediately.

When we separate the results by social vulnerability, the results suggest that people with higher level of vulnerability needs more time to evacuate than people with lower level of vulnerability. However, in our results, the differences between the evacuation curves are not statistically significant. In Figure 9, where we compare the aggregate survey responses with the evacuation responses categorized by social vulnerability level, we find that people at a medium level of vulnerability respond similarly to the aggregated values. Then, people with low and high vulnerability behave almost symmetrically around the average. Which in a more general application could be used to generate boundaries for the evacuation curves.

Insert Figure 9

To overcome the limitation of the no-significance in the difference between the evacuation curves more data need to be collected.

A limitation that arises when we apply a methodology such us ReTSVI, which relies on the construction of a social vulnerability index, is that we could not find studies that relate evacuation rates with social vulnerability for inundations that take less than an hour from the triggering to the flooding. In this study we used an SVI based on PCA to select the variables as proxy; however, this index was created and validated for post event

assessments. Therefore, this is a limitation that needs to be addressed before applying this framework.

Traditionally, the evacuation rate is calculated using one evacuation rate curve; therefore, ReTSVI seeks to overcomes this limitation by allowing the user to include social vulnerability. The user decides which social vulnerability index use and the evacuation curves for the levels of vulnerability. Here we provide an example using as a proxy a social vulnerability index for post disaster and evacuation curves that have not statistical significance. However, it still provides valuable information (Figure 8) of the implications of including social vulnerability that needs to be validated. For example, more vulnerable people, according to the SVI based on PCA and census data, live closer to the river where the inundation strikes earlier and harder, having less time to evacuate while at the same time they evacuate later. Additionally, social vulnerability seems to be less important as the EWS gets delayed."

**Comment 10 R2:** Citations in the conclusions are not necessary.

**Response comment 10:** We have deleted the references from the conclusions

**Comment 11 R2:** What's quality of dwelling materials expressed in line 6 of page 18?

**Response comment 11:** This refers to the quality of the house which were used in the older versions of the SVI. However, as this work progressed we decided to exclude variables related to physical vulnerability from this index, because that should be account into a physical vulnerability study that evaluates the risk of failure and then it should be combine to this or another version of SVI. However, we did not delete this sentence from the previous versions of this document and now we did it.

**Comment 12 R2:** Conclusions must give some critical final remarks about the "so what" of the paper for planners and risk managers. Limitations should be addressed in the discussion.

**Response comment 12:** We have rewritten this section and now it reads as follows:

[revised manuscript text omitted]

---

## Author Response (AR3)

Agricultural and Forestry Sciences

**UNIVERSIDAD DE LA FRONTERA**

*Avda. Francisco Salazar 01145, Casilla 54-D, 56-45-2325000, Temuco, Chile*

November 27, 2018

Dear Editor Dr. Paolo Tarolli

Thank you for considering our paper for a fourth revision. In this fourth submission, we submit a revised version of our paper with changes accepted, an abstract and in this file the response point by point to the comments and the changes to the article, and a revised version with changes that are not accepted. We hope that after this third revision we have clarified the reviewer comments that have centered mainly in the selection of the variables to construct the social vulnerability index. We did not intend to explain the factors that model social vulnerability neither to assume that a PCA based social vulnerability index is the definite way to describe social vulnerability in an evacuation process. However, we understand the concern of the reviewer and we addressed the comments in the following pages.

With all the best,

Dr. Marcelo Somos-Valenzuela

Corresponding author

**Comment Reviewer 1, third revision**

**Comment 1:** This manuscript continues to lack in demonstrating the choice and the validity of the variables chosen for the social vulnerability to evacuation. The authors argue that are widely considered in the computation of the index per se but not for evacuation purposes. I would consider socio-demographic characteristics rather than economic ones. How can income burden the capacity to evacuate?

**Response Comment 1:**

In the case of floods, we found few studies that seek to understand the relationship between evacuation and socioeconomic variables. For example, Henry et al., (2017) analyzed the relationship between income disparity and disaster information collection, and the resulting impacts on peoples' vulnerability after the 2011 Chao Phraya River Flood in Thailand. They found that among different demographic and socioeconomic variables, income was the strongest predictor of the population's decision to evacuate during the flood ($p<0.01$). They concluded that "…among those respondents affected by the flood, it could be seen that lower-income respondents had a higher tendency not to evacuate their homes" (page 5).

Furthermore, Medina & Moraca (2016) conducted a study to identify factors that influence the decision to evacuate upon flood warning by authorities  in the province of Bukidnon, Philippines.  They found that household income, measured as poverty, was a significant factor to explain whether families will evacuate upon advices by local authorities. According to Henry et al., (2017) household income is linked to different levels of access and use of information during flood evacuation and that could be one of mechanisms how income influences the decision making process to evacuate. They said "lower-income respondents tended to also utilize lower technology modes, such as radios and loudspeakers, in contrast to the internet-based modes used by higher-income respondents. Lower-income respondents also tended to be less aware of the government

hotline…" (page 9).  Moreover, research in social psychology and education show that wealthier families are more likely to take informed decisions because they have better access to information and more experience with choices and alternatives (Levin, 1998). Instead, low-income families have less access to information and are less likely to use official channels to make decisions (Ladd, 2002).

Therefore, we modified the document as follow:

In page 3 lines 25 to 29 the document originally said:

"Huang, Lindell, & Prater (2016) analyzed 49 studies conducted since 1991 linking evacuations to hurricane warnings and they concluded that demographic variables have a

minor or inconsistent impact on household evacuations. In the case of floods, however, we have not found any studies that link demographic and socioeconomic variables to the evacuation process."

Now it says:

"Huang, Lindell, & Prater (2016) analyzed 49 studies conducted since 1991 linking evacuations to hurricane warnings and they concluded that demographic variables have a minor or inconsistent impact on household evacuations. In the case of floods, however, we found few studies that seek to understand the relationship between evacuation and socioeconomic variables. For example, Henry et al., (2017) analyzed the relationship between income disparity and disaster information collection, and the resulting impacts on peoples' vulnerability after the 2011 Chao Phraya River Flood in Thailand. They found that among different demographic and socioeconomic variables, income was the strongest predictor of the population's decision to evacuate during the flood (p<0.01). They concluded that "…among those respondents affected by the flood, it could be seen that lower-income respondents had a higher tendency not to evacuate their homes." Furthermore, Medina & Moraca (2016) conducted a study to identify factors that influence the decision to evacuate upon flood warning by authorities  in the province of Bukidnon, Philippines.  They found that household income, measured as poverty, was a significant factor to explain whether families will evacuate upon advices by local authorities.

**Comment 2:** The capacity of people to evacuate less or faster is related to mobility factors, family interactions, community organization and information, distance from security points, etc. This makes me still skeptical in consideration of these variables and the outcomes of the current manuscript. Please justify it strongly. As you are proposing a new index, the choice and justification of the variables are essential.

**Response Comment 2:**

What the reviewer is pointing out here is the ability of the people to reach a safer area when the decision of evacuate has been taken. We did not use the variables in that context. We used them only to determine how social vulnerability affects the time when people decide to leave. As we mentioned in the comment 1 (above), there are not too many studies that seek to understand the relationship between the moment in which people decide to evacuate a river flood prone area and the population's demographic and socioeconomic characteristics. However, the literature found shows similar variables than this study. For example, Lim et al., (2016) pointed out "…that evacuation decision can be determined by a combination of household characteristics and capacities to cope with flood and hazard-related factors. Significant factors to evacuation decision include household characteristics (gender, educational level, presence of children less than or equal to 10 years old, and number of years living in the residence), capacity-related factors (house ownership, number of floors, type of house material), and hazard-related ones (distance from source of flood, level of flood damage, and source of warning)". Furthermore, Medina & Moraca (2016) found that "college education, having children below 5 years in the household, poverty, and depth of flood experienced positively influences a family's decision to evacuate upon flood warning in the study area". Finally, Henry et al. (2017) concluded that household income is the most relevant predictor, among other demographic and socioeconomic variables, to explain why people evacuate during flood in the 2011 Chao Phraya River Flood in Thailand.

The following table shows those variables that studies cited here have found relevant to explain whether population evacuate or stay at home during a flood evacuation (left side) and the variables used by this study (right side).

| Previous studies on river flood evacuation | Current study (RETSVI) |
| --- | --- |
| Gender | Women |
| Educational level | Population with primary education |
| | Population with college education |
| | Illiterate population |

| Presence of children | Children less than 1 year old |
|---|---|
| Poverty | Population with health insurance |
| | Informal settlement |
| | Household without electricity |
| | Indigenous population |
| Household income | Household with 5 or more rooms |
| | Population with "white collar jobs" |
| | Jobs in the manufacturing sector |
| | Jobs in the commerce sector |
| | Jobs in the construction sector |
| | Adult population divorced |
| House ownership, number of floors, type of house material | House rented |
| | Independent houses |
| | *Population with disabilities |
| | *Population older than 65 years old |

* These variables do not appear in studies of flood evacuation, (Henry et al., 2017; Lim et al., 2016; Medina & Moraca, 2016), but people with disabilities and elderly are more vulnerable according of the literature of social vulnerability post disaster. We decided to add them as part of the social vulnerability index because both variables affect the mobility and the time of reaction during a flood evacuation.

Although the variables that we selected to construct the social vulnerability index coincided with previous studies on evacuation to river flood events, it is important to mention again that the goal of this paper is not to prescribe a social vulnerability index within the ReSTVI methodology.

"The objective of this work is to propose a conceptual model 'The Response Time by Social Vulnerability Index (ReTSVI)' methodology that allows for the inclusion of social vulnerability into the traditional evacuation/mobilization models and it moves away from

traditional methods that combined social vulnerability and hazard magnitude by ranking in a matrix system that results in qualitative assessment." (page 6, lines 2 -6).

**Comment3:** Also, I would recommend adding the response 2 to reviewer 2 in the manuscript.

**Response Comment 3:** We include the response 2 to reviewer 2 from the second revision in the discussion section Page 13 Line 29 after "Insert Figure 9".

**Minor issues:**

**Minor issues 1:** Line 20-21 (abstract) I suggest to rephrase the sentence, it sounds odd.

**Response minor issue 1:**

The paper originally said: "The result of example of application have no statistical significance, which should be considered in a real case of application. Using a methodology such as ReTSVI could allow to combine social and physical vulnerability in a qualitative framework for evacuation, although, first more research needs to be done to understand the socioeconomic variables that explain the differences in evacuation rate. "

After the professional proofreader edition the paper now says: "The result of the application example has no statistical significance, which should be considered in a real case of application. Using a methodology such as ReTSVI could make it possible to combine social and physical vulnerability in a qualitative framework for evacuation, although more research is needed to understand the socioeconomic variables that explain the differences in evacuation rate."

**Minor issue 2:** In the abstract and conclusion, it is still mentioned the physical vulnerability. Do the authors include physical components in addition to the social ones?

**Response minor issue 2:**

In the conclusion we wrote "the physical vulnerability or the characteristics of an inundation event"

In the abstract we wrote "Using a methodology such as ReTSVI could allow to combine social and physical vulnerability in a qualitative framework for evacuation"

Yes, we include a physical model that describe the characteristics of an inundation Page 8, which provides the exposure and time to respond as well as the intensity of the flood although the intensity is not used in ReTSVI. Finally, knowing the intensity of the flood and the number of people that did not evacuate, we can estimate the number of people that can perish on a flood, which is not part of this study, but it can be one of the applications of this methodology.

**Minor issue 3:** I would recommend adding to the manuscript the results of the Bartlett's Sphericity test.

**Response minor issue 3:** We added it as Table 2 and renumbered the following tables.

**Minor issue 4:** Check carefully through the manuscript verbs tense consistency

**Response minor issue 4:** We sent the document for a professional proofreader (native speaker) to review and address the problems with the language.

**Minor issue 5:** Can you please add the total variance obtained from the PCA (that I feel is quite low)? What about the minimum value, the maximum one of the SVI? Can you please add them to the text?

**Response minor issue 5:** We already included the explained variance from the PCA, 57 %, in Table 3 which was Table 2 in the previous versions of this paper.

For the SVI values we repeated here our answer from Revision 1, reviewer 2 comment 21:

For the classification, we used three quantiles as it is shown in the figure below. The maximum value is 1.365, the minimum is -1.3425, the mean is 0.03, and the standard deviation is 0.4367

[Figure]

Figure 1 comments: Social Vulnerability Index statistics calculated in ArcGIS

[Figure]

Figure comment 21, revision 1, reviewer 2: Social Vulnerability Index classification calculated in ArcGIS.

The proportion of high, medium and low vulnerability blocks within the inundation zone is 15%, 35 %, and 50% respectively.

Therefore, to answer the reviewer comment we added the percentage of the variance explained by the PCA components selected in the second paragraph of section 3.2.1. And we added, "The resulted SVI ranges from -1.3424 to 1.365 with a mean of 0.03 and a standard deviation is 0.4367." After Table 2 and 3.

**Minor issue 6:** In addressing the limitations of the study, the authors said that there is a need for more data. Can you please define which ones and for which purposes?

**Response minor issue 6:** The limitation is explained in the same paragraph which relates to the lack of statically significance in the differences found. However, we added again this information, so this is even clearer for the readers.

[revised manuscript text omitted]

---

## Author Response (AR4)

Agricultural and Forestry Sciences

UNIVERSIDAD DE LA FRONTERA

*Avda. Francisco Salazar 01145, Casilla 54-D, 56-45-2325000, Temuco, Chile*

December 17, 2018

Dear Editor Dr. Paolo Tarolli

Thank you for considering our paper for a fifth revision. In this fifth submission, we submit a revised version of our paper with changes accepted, an abstract and in this file the response to the minor comment and the changes to the article, and a revised version with changes that are not accepted.

With all the best,

Dr. Marcelo Somos-Valenzuela

Corresponding author

**Minor Comment Reviewer 1, fourth revision**

I am suggesting a minor revision because it would be interesting to have some policy/managerial recommendations in light of the results obtained in order to translate the effort made with this research into a real emergency situation.

**Response Minor Comment 1:**

In order to respond to the reviewer suggestion, at the end of the discussion section we added

[revised manuscript text omitted]

---

## Author Response (AR5)

Agricultural and Forestry Sciences

**UNIVERSIDAD DE LA FRONTERA**

*Avda. Francisco Salazar 01145, Casilla 54-D, 56-45-2325000, Temuco, Chile*

January 14, 2019

Dear Editor Dr. Paolo Tarolli

Thank you for accepting with minor changes our paper for publication in NHESS. In the following page we address the comment you have made.

With all the best,

Dr. Marcelo Somos-Valenzuela

Corresponding author

**Minor Comment Editor**

I would suggest an improvement of the fig. 2 adding also a legend of colors; I printed it, it is difficult to read, also understand the colors range. In general, I would suggest a double check on the DPI of all the figures, in order to make these ready for high-quality online html version of the article.

**Response Minor Comment Editor:**

We have improved the quality of the figure and included a legend with colors. Also, we have check the quality of the figure to ensure that they are ready for publication.

[revised manuscript text omitted]